# Influence of age and biological sex on sweating in children exercising in warm and hot environments with comparison to adults

Thomas H. Topham[1] , James W. Smallcombe[2], Harry A. Brown[1], Brad Clark[1], Andrew P. Woodward[3,4], Richard D. Telford[1], Ollie Jay[2] and Julien D. Périard[1]

[1]*Research Institute for Sport and Exercise (UCRISE), University of Canberra, Bruce, ACT, Australia*
[2]*Heat and Health Research Incubator, Faculty of Medicine and Health, The University of Sydney, Sydney, NSW, Australia*
[3]*Faculty of Health, University of Canberra, Bruce, ACT, Australia*
[4]*Precision One Health Initiative, College of Veterinary Medicine, University of Georgia, Athens, GA, USA*

Handling Editors: Paul Greenhaff & Zachary Schlader

The peer review history is available in the Supporting Information section of this article (https://doi.org/10.1113/JP290323#support-information-section).

**Thomas Topham** completed a PhD at the University of Canberra Research Institute for Sport and Exercise and is currently a Performance Scientist at the Victorian Institute of Sport, Australia. He has a particular interest in sex-, fitness- and body composition-based differences in the physiological response to exercise and heat stress. **James Smallcombe** is a Senior Research Fellow at the Heat and Health Research Centre, University of Sydney, and Theme Lead for Maternal and Child Health. His research spans the lifespan, focusing on improving human resilience to extreme heat. He has particular expertise in paediatric and occupational heat stress and in protecting vulnerable populations during physical exertion in hot environments.

T. H. Topham & J. W. Smallcombe are co-first authors.

**Abstract figure legend** Local sweating response and sweating onset of boys, girls, adult females and adult males during treadmill walking at a fixed metabolic heat production per body surface area (300 W·m$^{-2}$) in warm and hot environments. Bayesian hierarchical modelling showed no meaningful sex or age differences in local sweating responses or onset threshold time in children when compared with adults.

**Abstract** Age and biological sex have been suggested to affect sweating during exercise, but studies that have compared the local sweating response between boys and girls of different ages are limited. We standardized exercise intensity to the rate of metabolic heat production per body surface area and compared local sweat rate and the sweating onset threshold between boys, girls, adult males and adult females using Bayesian hierarchical modelling. Sixty-one children (29 females; 10–16 years old) and 20 adults (9 females; 19–37 years old) walked for 45 min at a fixed rate of metabolic heat production per body surface area (300 W·m$^{-2}$) in WARM (30°C, 40% relative humidity) and/or HOT (40°C, 30% relative humidity) environments. Differences in local sweat rate between boys and girls were $\leq$0.03 mg·cm$^{-2}$·min$^{-1}$ (90% credible intervals: −0.14, 0.08) in WARM and $\leq$0.08 mg·cm$^{-2}$·min$^{-1}$ (−0.05, 0.23) in HOT, indicating no meaningful sex differences in children. In HOT, the local sweat rate of the back was 1.17 mg·cm$^{-2}$·min$^{-1}$ (1.05, 1.28) for boys, 1.09 mg·cm$^{-2}$·min$^{-1}$ (0.76, 1.41) for adult males, 1.09 mg·cm$^{-2}$·min$^{-1}$ (0.99, 1.19) for girls and 1.01 mg·cm$^{-2}$·min$^{-1}$ (0.72, 1.27) for adult females. Sweating onset threshold times differed by $\leq$0.6 min (−0.2, 1.4) between sexes at the back in WARM. An increase in age of 4 years was associated with a 1.1 min (0.1, 2.0) and 0.8 min (0.1, 1.5) greater sweating onset threshold time of boys and girls, respectively. In conclusion, age and biological sex did not meaningfully influence the local sweating response and onset threshold of children and adults exercising at 300 W·m$^{-2}$.

(Received 15 October 2025; accepted after revision 11 March 2026; first published online 14 April 2026)

**Corresponding authors** Julien D. Périard: University of Canberra Research Institute for Sport and Exercise Science, Bruce, ACT, Australia. Email: julien.periard@canberra.edu.au
Ollie Jay: Thermal Ergonomics Laboratory, Heat and Health Research Centre, Faculty of Medicine and Health, University of Sydney, NSW, Australia. Email: ollie.jay@sydney.edu.au

**Key points**

- Age and biological sex have been suggested to influence sweating during exercise, but their effect might be confounded by exercise intensity prescription and the consequent sweating requirement between individuals of differing body size.
- By standardizing exercise intensity to the rate of metabolic heat production per body surface area (in metres squared), we elicited an equivalent evaporative requirement for heat balance and compared local sweat rate and the sweating onset threshold between boys, girls, adult males and adult females exercising in warm and hot conditions.
- Our findings demonstrate that biological sex did not meaningfully influence the local sweating response and onset threshold time of children exercising at 300 W·m$^{-2}$ in warm and hot environments.
- In a hot environment, older children had a greater local sweat rate compared with younger children, but this was not evident when comparing children and adults.

## Introduction

Sweating is an essential thermoregulatory response when exercising under heat stress, with the evaporation of sweat enabling skin cooling. Previous studies have suggested that children have an underdeveloped sweating response (i.e. lower maximal sweat rate) (Falk et al., 1992) and different regional sweat patterns (Arlegui et al., 2021; Inoue et al., 2009; Shibasaki, Inoue, & Kondo, 1997) when compared with adults. Moreover, biological sex and maturation have both been purported to affect the sudomotor response to exercise (Amano et al., 2022; Arlegui et al., 2021) owing to physiological changes that occur with male puberty (i.e. increased testosterone) and

a greater physical capacity of adult male sweat glands, when compared with females (Gagnon & Kenny, 2012) and prepubertal boys (Inoue et al., 2009). However, few studies have directly examined sudomotor responses across childhood, especially in comparison to adults, and a previous review highlighted the need for more data to improve understanding in this area (Smith, 2019).

Previous research that has investigated the sudomotor response to exercise in children has often prescribed exercise based on an intensity relative to aerobic fitness [i.e. percentage of peak oxygen uptake ($\dot{V}_{O_2peak}$)] (Amano et al., 2022; Arlegui et al., 2021) or an absolute work rate (Haymes et al., 1974; Haymes et al., 1975). However, both protocols can create differences in the rate of metabolic heat production ($\dot{H}_{prod}$) and the consequent evaporative requirement for heat balance ($E_{req}$) between participants of differing aerobic fitness and body size (Cramer & Jay, 2014). Several studies have demonstrated that whole-body sweat rate (WBSR) is primarily driven by $E_{req}$ in watts, such that any differences in $E_{req}$ between individuals as a product of the exercise intensity prescription must be standardized before comparing WBSR (Cramer & Jay, 2014, 2015; Gagnon et al., 2013). For example, a greater WBSR has been demonstrated with advanced pubertal stage in boys cycling at 50% $\dot{V}_{O_2peak}$ in 42°C (Falk et al., 1992), yet this has not been observed in girls when compared with adult women (Rivera-Brown et al., 2006), which suggests a male hormonal effect on the sweating response (Inoue et al., 2004). However, exercise intensity in these paediatric studies was prescribed at a percentage of $\dot{V}_{O_2peak}$, which fails to account for the potential confounding effect of differences in $E_{req}$ (in watts) between participants (Cramer & Jay, 2014, 2016). As such, the greater WBSR with advanced pubertal stage reported in boys might be the product of differences in $E_{req}$, rather than age or maturational status. Accordingly, we recently demonstrated that WBSR is greater in adults than in children when exercising at intensities scaled to fitness ($\dot{V}_{O_2peak}$), body mass (in watts per kilogram) and body surface area (BSA; in watts per metre squared) (Smallcombe et al. 2025).

Studies that measure steady-state sudomotor activity over a fixed skin surface area [i.e. local sweat rate (LSR)] are required to normalize the exercise intensity to the BSA of an individual (Cramer & Jay, 2014, 2015). This approach has been used successfully with adult participants to address the confounding effect of differences in body size between participants and allow for the effect of the experimental variable (e.g. biological sex, aerobic fitness) to be isolated (Cramer & Jay, 2014; Jay et al., 2011; Ravanelli et al., 2017; Smoljanić et al., 2014). A previous study in adults that standardized exercise intensity at a fixed $\dot{H}_{prod}$ per BSA ($\leq 250$ W·m$^{-2}$) reported that biological sex did not influence LSR or whole-body evaporative heat loss (in W·m$^2$) when exercising in 40°C

(Gagnon & Kenny, 2012). However, females presented a lower LSR on the back, chest and forearm, hence a diminished whole-body evaporative heat loss ($\sim$40 W·m$^{-2}$) during exercise at 300 W·m$^{-2}$ (Gagnon & Kenny, 2012). This observation suggests that females have a lower sweating capacity and therefore reach maximum sweat rate at a lower $E_{req}$ than males. Differences in sweat distribution have been reported between adult males and females, with males producing a higher proportion of sweat from the torso, when compared with females (Smith & Havenith, 2012). Taken together, these findings suggest that biological sex might influence both maximum sweating capacity and sweat distribution in adults, which is not evident in less thermally stressful environments. If these sex-based differences in sweating are consistent in children, we expect boys to have a higher sweating capacity compared with girls in conditions of greater thermal strain. However, whether biological sex has a similar effect within children, and when compared with adults, is unclear.

Following our recent investigation of WBSR responses in children and adults (Smallcombe et al., 2025), the present study extends that work by providing the first comprehensive examination of age- and biological sex-associated differences in local sweat rate and sweating onset threshold, while controlling for differences in body size via standardization of metabolic heat production. The primary aim of this study was to investigate the influence of age and biological sex on LSR of children aged 10–16 years exercising in warm and hot environments at a standardized $\dot{H}_{prod}$ per BSA (in W·m$^2$). The secondary aim was to compare the sudomotor response of children with adults exercising at the same target $\dot{H}_{prod}$ to identify differences between children and adults in sweating.

## Methods

### Ethical approval

Ethical approval was obtained from the Research Ethics committees of University of Canberra (UC) (#20 204 538) and University of Sydney (USyd) (HREC no. 2016/983), and the trials were conducted in accordance with the *Declaration of Helsinki*, except for registration in a database.

### Participants

Thirty-two boys and 29 girls volunteered for this study (Table 1). Seventeen children [4 males (M) and 13 females (F)] completed both 30°C, 40% relative humidity (RH) (WARM) and 40°C, 30% RH (HOT) trials. In total, 44 (20 M and 24 F) trials were completed in WARM and 34 (17 M and 18 F) trials were completed in HOT. To permit

**Table 1. Physical and physiological characteristics of boys, girls, adult males and females.**

| Characteristic | Boys | Girls | Adult males | Adult females |
| --- | --- | --- | --- | --- |
| Age (years) | 13.5 (1.4) | 13.3 (1.9) | 26.8 (5.3) | 28.5 (3.8) |
| Height (m) | 1.65 (0.12) | 1.59 (0.09) | 1.81 (0.06) | 1.64 (0.04) |
| Body mass (kg) | 55.4 (14.8) | 51.0 (9.1) | 77.4 (7.0) | 62.4 (10.7) |
| BSA (m$^2$) | 1.59 (0.24) | 1.50 (0.17) | 1.97 (0.11) | 1.67 (0.41) |
| BSA/Body mass (cm$^2 \cdot$kg$^{-1}$) | 296 (33) | 299 (23) | 255 (11) | 272 (22) |
| $\dot{V}_{O_2 peak}$ (mL$\cdot$kg$^{-1} \cdot$min$^{-1}$) | 53 (10) | 46 (7) | 56 (5) | 47 (7) |
| Tanner stage (pre/mid/late) | 1/23/4 | 2/26/1 | – | – |

Data are expressed as the mean (SD). Boys, $n = 32$; girls, $n = 29$; adult males, $n = 11$; adult females, $n = 9$. Tanner stages are as follows: 1, prepubertal; 2, 3 and 4, midpubertal; and 5, late-pubertal. Abbreviations: BSA, body surface area; $\dot{V}_{O_2 peak}$, peak oxygen consumption.

a comparison between children and adults, 11 adult males and 9 adult females volunteered for this study (Table 1). Eleven (7 M and 4 F) adults completed trials in both conditions. In total, 17 (11 M and 6 F) adult trials were completed in WARM and 15 (7 M and 8 F) adult trials in HOT.

The legal guardians of the children provided written consent, each child gave written assent, and adult participants provided written consent, prior to participation. All participants were deemed free from metabolic and cardiovascular diseases following completion of a pre-exercise screening questionnaire (Exercise and Sports Science Australia Pre Exercise Screening Tool). Biological maturation was determined via self-assessed Tanner staging (Tanner, 1962) in 57 participants. Data were collected at two sites: 35 children (16 M and 19 F) and 15 adults (7 M and 8 F) visited the University of Canberra (UC), and 26 children (16 M and 10 F) and 5 adults (4 M and 1 F) visited the University of Sydney (USyd).

### Preliminary visit

Participants attended the laboratory for a preliminary visit that was standardized between study sites and included anthropometric measurements and an exercise test. Height and body mass were measured with a wall-mounted stadiometer (UC: Seca 220, Germany; USyd: Hotltain, Crosswell, UK) and digital scale (UC: KW Industrial platform scales, Atweigh, Victoria, Australia; USyd: Mettler Toledo, Germany), respectively. The BSA was calculated using the Du Bois & Du Bois (1916) equation, which is deemed appropriate for use in children and adolescents with a BSA of $>0.45$ m$^2$ (Haycock et al., 1978):

$$\text{BSA} = 0.202 \times \text{body mass}^{0.425} \times \text{height}^{0.72}.$$

The preliminary exercise test consisted of submaximal and maximal portions performed in temperate conditions (20°C–24°C, 40%–60% RH) on a motorized treadmill. The submaximal portion determined the treadmill gradient required to elicit the target $\dot{H}_{prod}$ for the experimental trials (Cramer & Jay, 2014). Expired gases were measured via indirect calorimetry (UC: ParvoMedics, UT, USA; USyd: Quark CPET, COSMED, Rome, Italy) during five 4-min stages of walking at a constant speed (range: 5.0–6.0 km$\cdot$h$^{-1}$ based on participant height and comfort). The treadmill incline was set to 2% for the first stage and increased by 2% per stage thereafter. The corresponding oxygen uptake ($\dot{V}_{O_2}$) was defined as the mean value for the final 1 min of each stage. Following a 10-min rest period, $\dot{V}_{O_2 peak}$ was determined using a maximal ramp test. Participants ran at a self-selected speed for 1 min at 0% incline. Thereafter, the treadmill incline was increased by 1% every 1 min until volitional exhaustion. The $\dot{V}_{O_2 peak}$ was defined as the highest 30 s mean $\dot{V}_{O_2}$.

### Experimental trial

Participants were asked to refrain from strenuous exercise for 24 h prior to the experimental trial. Upon arrival at the laboratory ($\sim$30 min prior to testing), a urine sample was provided, and hydration status was assessed via the measurement of urine specific gravity using a hand-held refractometer (UC and USyd: PEN-Urine S.G., Atago Co. Ltd, Tokyo, Japan). Pretrial euhydration was verified with a urine specific gravity of $<1.025$ (Kenefick & Cheuvront, 2012). If hydration was required, participants consumed 5 mL per kg of body mass of water, served at room temperature. Clothing to be worn in the trial (running shorts, singlet and socks) were weighed on calibrated digital scales (UC: Anko, Mulgrave, Melbourne, Victoria, Australia; USyd: Mettler Toledo, Germany). Participants then changed into the clothing, and initial body mass measurements were taken. Nude body mass was calculated as clothed body mass minus clothing mass (Cheuvront & Kenefick, 2017). Next, participants were instrumented with skin temperature sensors before resting in the seated position for 5 min to obtain baseline measurements.

Participants then sat for 10 min in a climate-controlled environmental chamber set to WARM or HOT conditions before walking on the motorized treadmill for 45 min at the same submaximal speed as their preliminary trial, with the treadmill incline set to elicit a $\dot{H}_{prod}$ of 300 W·m$^{-2}$. The work rate of each trial was verified by indirect calorimetry measured during the first 20 min of exercise. Gastro-intestinal temperature ($T_{gi}$) and skin temperature were measured continuously. All trials were completed without artificial convection (i.e. fanning) and fluid ingestion. Postexercise clothed body mass was recorded immediately upon cessation of exercise, and clothing mass was then measured separately.

## Measurements

The $T_{gi}$ was measured using a telemetric sensor pill and remote monitoring system (e-Celsius, BodyCap, Caen, France). Participants ingested the pill ˜8 h before the experimental trial, and $T_{gi}$ data were collected at a sampling rate of 15 s. Whole-body sweat loss was estimated by measuring changes in body mass from pre- to postexercise. Body mass was measured in duplicate, and whole-body sweat loss was corrected for respiratory and metabolic mass loss (Mitchell et al., 1972). The WBSR (in grams per hour) was calculated by dividing whole-body sweat loss by exercise time.

The LSR was measured using ventilated sweat capsules affixed to the dorsal forearm (∼5 cm distal to the elbow; LSR$_{arm}$) and the upper back (∼2 cm lateral to the scapula; LSR$_{back}$) with surgical tape (Transpore; 3M, North Ryde, NSW, Australia). Anhydrous air was passed through each capsule at a continuous flow rate of 0.5 L·min$^{-1}$. The temperature and humidity of effluent air from the capsules were measured using a factory-calibrated capacitance hygrometer (HMT333, Vaisala; Vantaa, Finland). The LSR was calculated as the product of flow rate and the difference in absolute humidity between effluent and influent air, normalized to the skin surface area beneath the capsule (4.0 cm$^2$) and expressed as milligrams per centimetre squared per minute. The LSR data were collected at a sampling rate of 5 s, and 1 min averages were calculated for sweating onset threshold analysis and recorded every 5 min for LSR analysis.

Skin temperature was measured at four sites on the right side of the body using iButtons (Maxim Integrated Products, San Jose, CA, USA) affixed to the skin surface with adhesive tape. Mean skin temperature ($T_{sk}$) was calculated using Ramanathan (1964) weighting coefficients: chest (30%), shoulder (30%), thigh (20%) and calf (20%). The $T_{sk}$ data were collected at a sampling rate of 5 s and averaged over 1 min at 5 min intervals during exercise.

## Heat balance calculations

The $\dot{H}_{prod}$ was calculated as the difference between metabolic energy production ($M$) and external work rate (Wk):

$$\dot{H}_{prod} = M - \mathrm{Wk} \; (\mathrm{W \cdot m^{-2}}) \qquad (1)$$

where $M$ was estimated by entering the mean $\dot{V}_{O_2}$ (in L·min$^{-1}$) and respiratory exchange ratio (RER) measured during exercise into the following equation (Nishi, 1981):

$$M = \dot{V}_{O_2} \times \frac{\left(\left(\frac{RER-0.7}{0.3}\right) \times e_c\right) + \left(\left(\frac{1.0-RER}{0.3}\right) \times e_f\right)}{60 \times \mathrm{BSA}}$$
$$\times 1000 \; (\mathrm{W \cdot m^{-2}}) \qquad (2)$$

where $e_c$ and $e_f$ represent the energy equivalent per litre of oxygen consumed (in L·min$^{-1}$) for carbohydrate (21.13 kJ) and fat (19.62 kJ), respectively.

The Wk on the treadmill was calculated as follows (Gibson et al., 1979):

$$\mathrm{Wk} = \frac{g_0 \times m \times v \times (i/100)}{\mathrm{BSA}} \; (\mathrm{W \cdot m^{-2}}) \qquad (3)$$

where $g_0$ represents gravitational acceleration (in m·s$^{-1}$), $m$ is body mass (in kg), $v$ is treadmill belt velocity (in m·s$^{-1}$), and $i$ is treadmill incline (as a percentage).

Sensible heat exchange at the skin surface was calculated as the sum of convective ($C$) and radiant ($R$) heat transfer. Convective heat exchange ($C$) was calculated as:

$$C = h_c \times (T_{sk} - T_a) \; (\mathrm{W \cdot m^{-2}}) \qquad (4)$$

where $T_a$ is ambient temperature (°C), and $h_c$ is the convective heat transfer coefficient for treadmill walking (Nishi & Gagge, 1970):

$$h_c = 6.51 \times v^{0.391} \; (\mathrm{W \cdot m^{-2} \cdot K^{-1}}) \qquad (5)$$

corrected for barometric pressure ($P_b$) by a factor of $(P_b/760)^{0.55}$ (Gagge & Nishi, 1977).

Radiant heat loss ($R$) was calculated as:

$$R = h_r \times (T_{sk} - T_a) \; (\mathrm{W \cdot m^{-2}}) \qquad (6)$$

where $h_r$ is the radiant heat transfer coefficient:

$$h_r = 4\varepsilon\sigma \times (\mathrm{BSA}_r/\mathrm{BSA})$$
$$\times ([T_{sk} + T_r]/2 + 273.15)^3 \; (\mathrm{W \cdot m^{-2} \cdot K^{-1}}) \quad (7)$$

where $\varepsilon$ is the area-weighted emissivity of the skin (0.95), $\sigma$ is the Stefan–Boltzmann constant (5.67 × 10$^{-8}$ W·m$^{-2}$ K$^{-4}$), BSA$_r$/BSA is the effective radiant surface area of the body (no denomination) for a walking person (0.72), and $T_r$ is the mean radiant temperature (in °C), assumed to be equal to $T_a$.

Respiratory heat losses via evaporation ($E_{res}$) and convection ($C_{res}$) were estimated by (Cramer & Jay, 2019):

$$C_{res} + E_{res} = 0.001516 \times M \times (28.56 + 0.641 \times P_a$$
$$- 0.885 \times T_a) + 0.00127 \times M \times (59.34 + 0.53$$
$$\times T_a - 11.63 \times P_a)(W \cdot m^{-2}) \quad (8)$$

where $P_a$ is the ambient vapour pressure (in kPa).

$E_{req}$ was calculated as:

$$E_{req} = \dot{H}_{prod} - (C + R + E_{res} + C_{res}) \, (W \cdot m^{-2}) \quad (9)$$

$E_{max}$ was calculated as:

$$E_{max} = \omega_{max} \left( P_{sk,s} - P_a \right)$$
$$/ \left[ R_{e,cl} + \left( [1/h_e \, f_{cl}] \right) \right] \, (W \cdot m^{-2}) \quad (10)$$

where $\omega_{max}$ is maximum skin wettedness (Gagge, 1937), set as 1.00 for all participants, $R_{e,cl}$ is the evaporative heat transfer resistance of the clothing ensemble (assumed to be 0.01 kPa·m$^{-2}$·W$^{-1}$); $f_{cl}$ is the clothing area factor (surface area of clothed body divided by surface area of nude body; assumed to be negligible); $h_e$ is the evaporative heat transfer coefficient (in W·m$^{-2}$·kPa$^{-1}$), which is the product of $h_c$ (equation 6) and the Lewis relationship (16.5 K·kPa$^{-1}$), corrected for $P_b$ $(760/P_b)^{0.45}$, and $P_{sk,s}$ is the saturated water vapour pressure (in kPa) at $T_{sk}$ derived from the Antoine equation (Osborne & Meyers, 1934).

## Statistical analysis

The WBSR was analyzed with a Bayesian linear hierarchical model. The predictor variables were the interaction of age (in years) × sex (male/female). Priors over parameters were set as weakly informative distributions (mean, SD): intercept 500 g·h$^{-1}$, 200; sex 0 g·h$^{-1}$, 200; and age 0 g·h$^{-1}$, 100. For $T_{gi}$, LSR$_{back}$ and LSR$_{arm}$, a Bayesian hierarchical generalized additive model (Pedersen et al., 2019) was implemented, with time, sex (male/female) and age (in years) as the predictor variables. This analysis method was chosen because the LSR and $T_{gi}$ by time relationships were non-linear (Mundo et al., 2022). A hierarchical generalized additive model implements penalized smoothing splines that allow the form of the relationship to be determined from the data (Wood, 2004). Penalization of the splines reduces the impact of overfitting to the data (Mundo et al., 2022). A normal prior distribution was chosen for the intercept (time point 0) with a mean of 37.0°C and SD of 1°C for $T_{gi}$ and a mean of 0.2 mg·cm$^{-2}$·min$^{-1}$ and SD of 0.2 mg·cm$^{-2}$·min$^{-1}$ for LSR, respectively.

Time of sweating onset threshold of the back and arm was determined using segmented non-linear regression, with age (in years) × sex (male/female) as the predictor variables. The model defined the time point at which the LSR–time relationship altered from linear to an asymptotic growth curve. The time of transition and model estimates for a particular age and sex were calculated within one analysis. Onset thresholds were calculated in WARM only, because sweating occurred prior to exercise commencing in HOT.

To evaluate the effect of age, we generated posterior predictions from the fitted model for children nominally aged 11 and 15 years, representing the grand mean age (13 years) plus and minus 2 years, respectively. These predictions were then compared to assess the influence of a difference in age of 4 years. To evaluate differences between adults and children, WBSR, LSR, $T_{gi}$ and onset threshold analyses were repeated with adults. Posterior mean predictions were calculated for adult males, adult females, boys and girls with their respective mean age. Models were implemented using the 'brms' package (Bürkner, 2017) using R (R Development Core Team, 2022) in RStudio v.4.2.1. Model convergence was assessed using the Rhat statistic and the effective sample size using Markov chain Monte Carlo sampling (Vehtari et al., 2021). Graphics were developed with the 'ggplot2' package (Wickham, 2016). Parameter estimates (differences) are reported as estimated means and 90% credible intervals, which represent a 90% probability that the true estimate is within the interval, given the evidence provided by the observed data. Probability of direction (Pd) statements, which express the probability that the value of the prediction is in the direction of the point estimate, were implemented using the 'BayestestR' package (Makowski et al., 2019).

## Results

### Heat exchange

Heat exchange for children and adults in WARM and HOT conditions is presented in Table 2. Mean $\dot{H}_{prod}$ (in W·m$^{-2}$) was similar between children and adults, varying between 297 and 305 W·m$^{-2}$. Mean $E_{req}/E_{max}$ was 1.3 in HOT and 1.1 in WARM. Adult males had a 78–114 and 106–141 W greater mean $E_{req}$ than boys, girls and adult females in WARM and HOT, respectively. The mean $E_{req}$ of boys, girls and adult females did not differ by >36 W.

### Local sweat rate

**WARM.** The difference in end-exercise LSR$_{back}$ between sexes (boys–girls) was −0.03 mg·cm$^{-2}$·min$^{-1}$ (−0.14, 0.08; Pd = 67%). An age difference of 4 years (from 11 to 15 years) was associated with a change in end-exercise LSR$_{back}$ of −0.08 mg·cm$^{-2}$·min$^{-1}$ (−0.25, 0.08; Pd = 79%; Fig. 1*A*) for boys and 0.07 mg·cm$^{-2}$·min$^{-1}$ (−0.05, 0.19; Pd = 83%; Fig. 1*A*) for girls. The mean difference in

end-exercise LSR$_{back}$ was 0.17 mg·cm$^{-2}$·min$^{-1}$ (−0.06, 0.40; Pd = 89%) between adult males and boys, and −0.05 mg·cm$^{-2}$·min$^{-1}$ (−0.33, 0.23; Pd = 60%) between adult females and girls (Table 3).

The difference in end-exercise LSR$_{arm}$ between sexes (boys–girls) was −0.00 mg·cm$^{-2}$·min$^{-1}$ (−0.08, 0.08; Pd = 55%). An age difference of 4 years (from 11 to 15 years) was associated with a change in end-exercise LSR$_{arm}$ of −0.13 mg·cm$^{-2}$·min$^{-1}$ (−0.25, 0.01; Pd = 96%; Fig. 1*B*) for boys and 0.04 mg·cm$^{-2}$·min$^{-1}$ (−0.05, 0.12; Pd = 77%; Fig. 1*B*) for girls. The mean difference in end-exercise LSR$_{arm}$ was 0.00 mg·cm$^{-2}$·min$^{-1}$ (−0.14, 0.15; Pd = 52%) between adult males and boys, and −0.07 mg·cm$^{-2}$·min$^{-1}$ (−0.25, 0.10; Pd = 76%) between adult females and girls (Table 3).

**HOT.** The difference in end-exercise LSR$_{back}$ between sexes (boys–girls) was 0.08 mg·cm$^{-2}$·min$^{-1}$ (−0.06, 0.23; Pd = 82%). An age difference of 4 years (from 11 to 15 years) was associated with a change in end-exercise LSR$_{back}$ of 0.21 mg·cm$^{-2}$·min$^{-1}$ (−0.07, 0.49; Pd = 93%; Fig. 2*A*) for boys and 0.12 mg·cm$^{-2}$·min$^{-1}$ (−0.13, 0.36; Pd = 83%; Fig. 2*A*) for girls. The mean difference in end-exercise LSR$_{back}$ was −0.08 (−0.42, 0.26; Pd = 66%) between adult males and boys, and −0.08 (−0.39, 0.20; Pd = 67%) between adult females and girls (Table 3).

The difference in end-exercise LSR$_{arm}$ between sexes (boys–girls) was 0.08 mg·cm$^{-2}$·min$^{-1}$ (−0.05, 0.20; Pd = 85%). An age difference of 4 years (from 11 to 15 years) was associated with a change in end-exercise LSR$_{arm}$ of 0.04 mg·cm$^{-2}$·min$^{-1}$ (−0.17, 0.24; Pd = 62%; Fig. 2*B*) for boys and 0.19 mg·cm$^{-2}$·min$^{-1}$ (−0.03, 0.35; Pd = 97%;

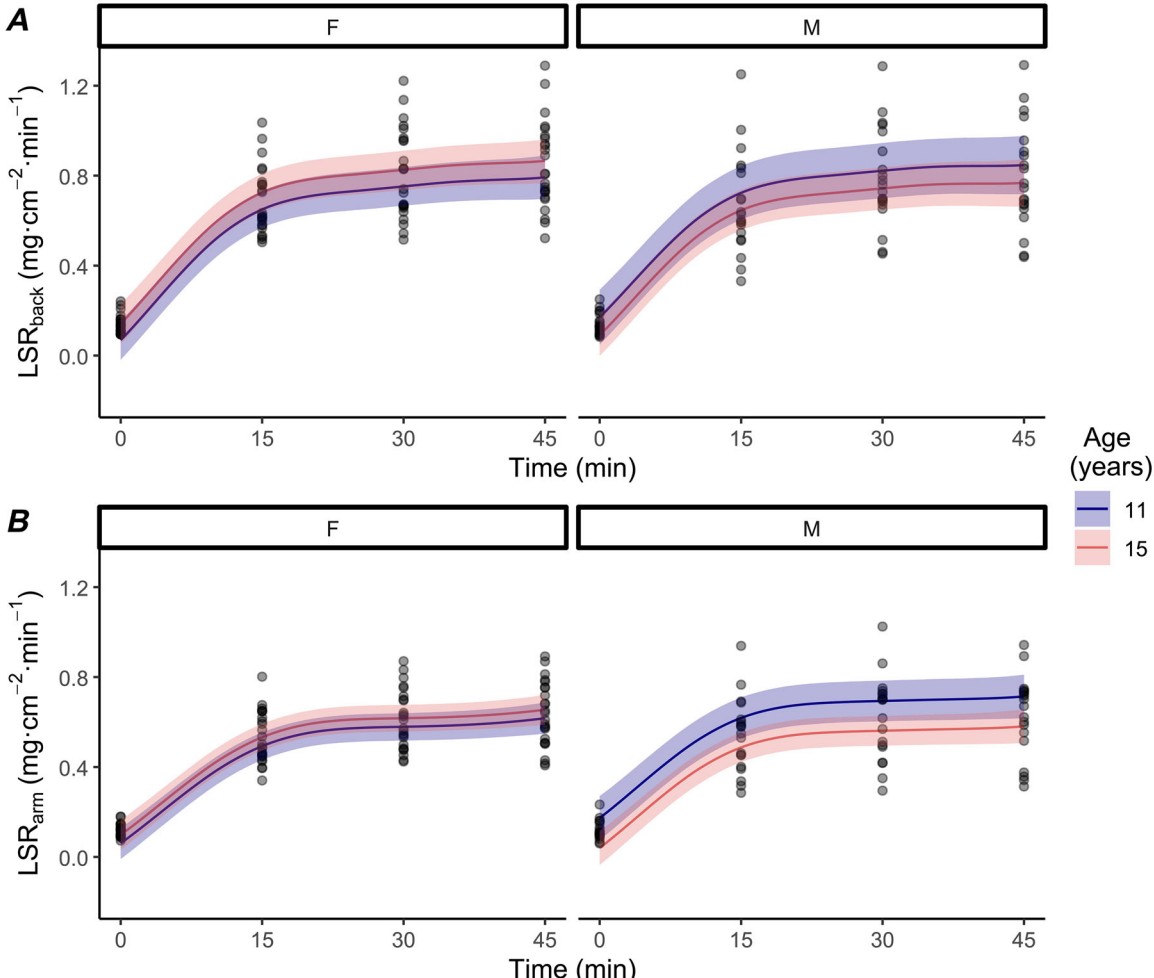

**Figure 1. Back local sweat rate and arm local sweat rate of girls and boys during exercise in WARM conditions**
Back local sweat rate (LSR$_{back}$; *A*) and arm local sweat rate (LSR$_{arm}$; *B*) of girls (F; *n* = 24) and boys (M; *n* = 20) during exercise in WARM conditions. Data were analyzed with a Bayesian hierarchical generalized additive model implemented with time, sex (male/female) and age (in years) as the predictor variables. Age is predicted as 11 and 15 years. Coloured lines are population predicted means with 90% credible intervals (ribbon). Points are individual participant data presented every 15 min during exercise.

Fig. 2*B*) for girls. The mean difference in end-exercise LSR$_{arm}$ was 0.07 (−0.12, 0.27; Pd = 75%) between adult males and boys, and 0.07 (−0.10, 0.24; Pd = 77%) between adult females and girls (Table 3).

## WBSR

**WARM.** Per-year changes in WBSR (slope) derived from posterior estimates indicate a weaker relationship in WARM (boys: 14 g·h$^{-1}$ per year [−5, 34], Pd = 88%; girls: 24 [8, 40] g·h$^{-1}$ per year, Pd = 99%) than HOT (boys: 64 g·h$^{-1}$ per year [39, 89], Pd >99%; girls: 53 g·h$^{-1}$ per year [29, 76], Pd >99%) with a <10 g·h$^{-1}$ per year difference between male and female slopes (Fig. 3*A* and *B*). The difference in WBSR between sexes (boys–girls) was −24 g·h$^{-1}$ (−73, 24; Pd = 80%; Fig. 3*A*). An age difference of 4 years (from 11 to 15 years) was associated with a change in WBSR of 57 g·h$^{-1}$ (−19, 135; Pd = 88%) for boys and 94 g·h$^{-1}$ (31, 159; Pd = 99%) for girls (Fig. 3*A*). There was a mean difference in WBSR of 148 g·h$^{-1}$ (57, 219; Pd = >99%) between adult males and boys, and −22 g·h$^{-1}$ (−67, 23; Pd = 78%) between adult females and girls (Table 3).

**HOT.** The difference in WBSR between sexes (boys–girls) was −22 g·h$^{-1}$ (−67, 23; Pd = 71%; Fig. 3*B*). An age difference of 4 years (from 11 to 15 years) was associated with a change in WBSR of 256 g·h$^{-1}$ (155, 357; Pd >99%) for boys and 211 g·h$^{-1}$ (117, 305; Pd >99%) for girls (Fig. 3*B*). There was a mean difference in WBSR of 85 g·h$^{-1}$ (−51, 219; Pd = 84%) between adult males and boys, and 14 g·h$^{-1}$ (−76, 104; Pd = 60%) between adult females and girls (Table 3).

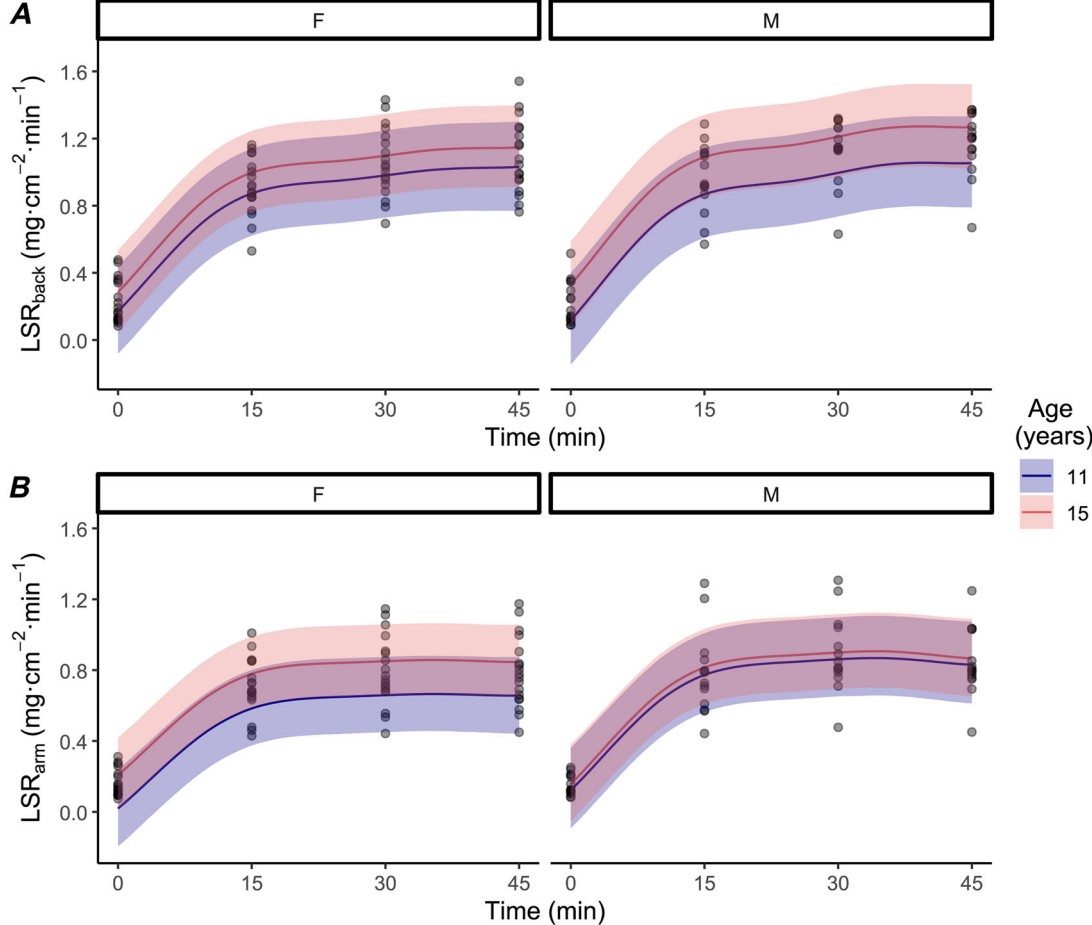

**Figure 2. Back local sweat rate and arm local sweat rate of girls and boys during exercise in HOT conditions**
Back local sweat rate (LSR$_{back}$; *A*) and arm local sweat rate (LSR$_{arm}$; *B*) of girls (F; *n* = 18) and boys (M; *n* = 17) during exercise in HOT conditions. Data were analyzed with a Bayesian hierarchical generalized additive model implemented with time, sex (male/female) and age (in years) as the predictor variables. Age is predicted as 11 and 15 years. Coloured lines are population predicted means with 90% credible intervals (ribbon). Points are individual participant data presented every 15 min during exercise. One boy did not have LSR$_{arm}$ measured (*n* = 16).

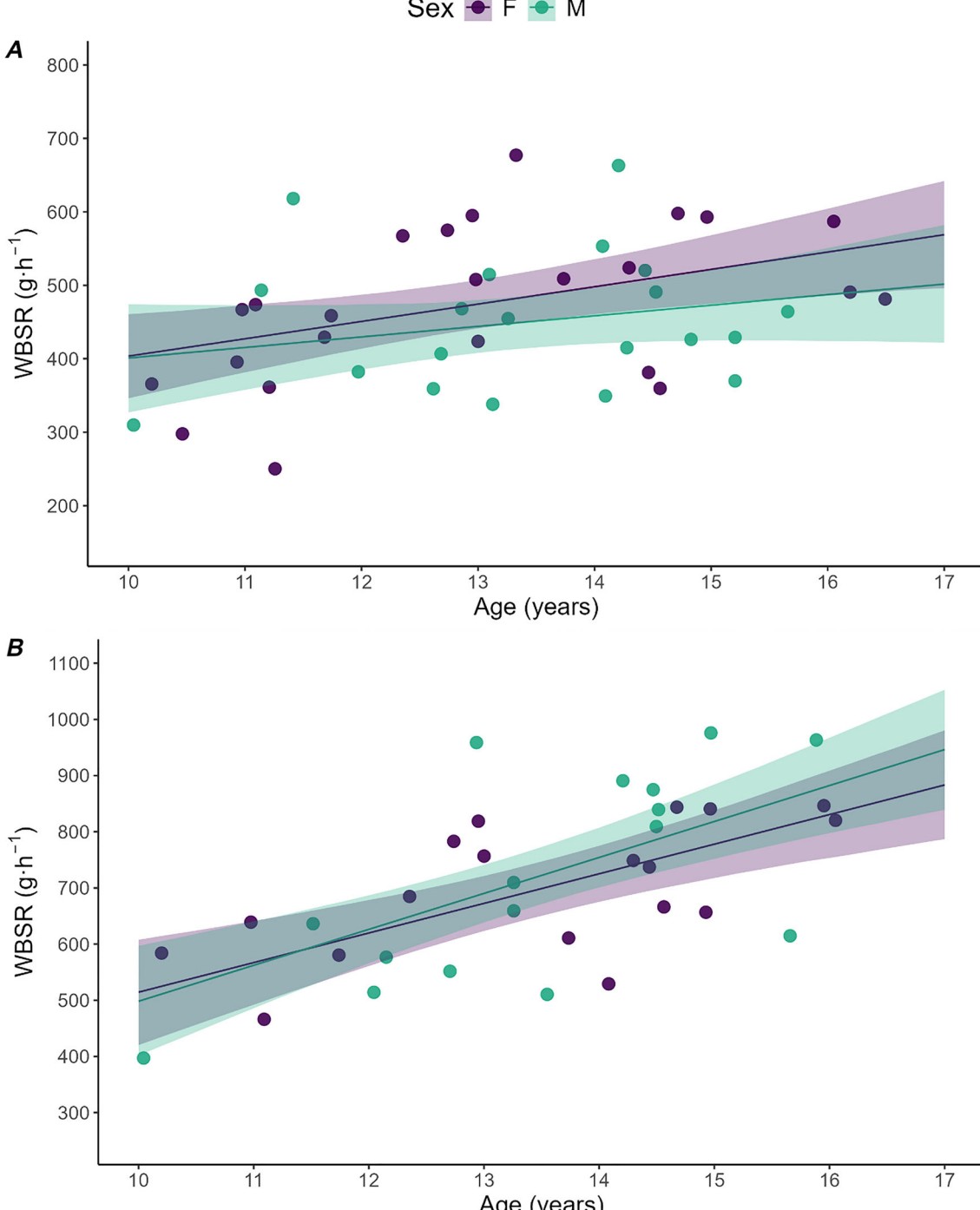

**Figure 3. Whole-body sweat rate of girls and boys during exercise in WARM and HOT conditions relative to age**
Whole-body sweat rate (WBSR) of girls (F) and boys (M) during exercise in WARM (*A*) and HOT (*B*) conditions relative to age (circles), with model estimates (line) and 90% credible intervals (ribbon). WARM: girls, *n* = 24; boys, *n* = 20; HOT: girls, *n* = 18; boys, *n* = 17. Slopes represent the change in hourly WBSR (in grams per hour) per year of age.

**Table 2. Heat exchange during exercise at a metabolic heat production per body surface area of 300 W·m⁻² in boys, girls, adult males and females in warm and hot conditions.**

| Parameter | WARM | | | | HOT | | | |
|---|---|---|---|---|---|---|---|---|
| | Boys | Girls | Adult males | Adult females | Boys | Girls | Adult males | Adult females |
| $\dot{H}_{prod}$ (W·m⁻²) | 297 (10) | 301 (9) | 305 (6) | 297 (15) | 299 (9) | 300 (11) | 305 (7) | 299 (11) |
| $\dot{H}_{prod}$ (W) | 473 (72) | 456 (58) | 603 (40) | 495 (56) | 486 (79) | 460 (49) | 598 (47) | 492 (45) |
| $\dot{H}_{prod}$ (W·kg⁻¹) | 8.8 (1.1) | 8.9 (0.7) | 7.7 (0.3) | 8.1 (0.6) | 8.6 (1.2) | 8.8 (0.7) | 7.9 (0.3) | 8.2 (0.5) |
| $E_{req}$ (W) | 384 (63) | 375 (53) | 489 (40) | 411 (57) | 503 (86) | 476 (53) | 617 (54) | 511 (43) |
| $C + R$ (W·m⁻²) | 34 (8) | 28 (11) | 36 (9) | 28 (8) | −28 (6) | −28 (6) | −29 (7) | −31 (5) |
| $E_{req}$ (W·m⁻²) | 241 (10) | 248 (13) | 247 (12) | 246 (20) | 309 (9) | 310 (12) | 315 (12) | 310 (8) |
| $E_{max}$ (W·m⁻²) | 223 (11) | 215 (15) | 223 (14) | 219 (20) | 235 (11) | 238 (13) | 240 (9) | 248 (11) |
| $E_{req}/E_{max}$ (ND) | 1.09 (0.08) | 1.16 (0.13) | 1.12 (0.12) | 1.14 (0.20) | 1.32 (0.09) | 1.31 (0.09) | 1.31 (0.09) | 1.25 (0.05) |

Values are expressed as the mean (SD). WARM: boys, $n = 20$; girls, $n = 17$; girls, $n = 18$; adult males, $n = 7$; adult females, $n = 6$; HOT: boys, $n = 17$; girls, $n = 18$; adult males, $n = 7$; adult females, $n = 11$; adult males, $n = 24$; $n = 8$. Abbreviations: C, convective heat exchange; $E_{max}$, maximum rate of evaporation; $E_{req}$, rate of evaporation required for heat balance; $\dot{H}_{prod}$, metabolic heat production; ND, no denomination; R, radiative heat exchange. Positive values for $C + R$ indicate heat loss, whereas negative values indicate heat gain.

## Sweating onset threshold

**WARM.** Model estimates indicate a weak positive relationship between onset threshold time and age in both sexes for back (girls: slope = 0.2 min per year [0, 0.4], Pd = 98%; boys: slope = 0.3 min per year [0, 0.5], Pd = 96%) and arm (girls: slope = 0.2 min per year [0.1, 0.4], Pd = 98%; boys: slope = 0.2 min per year [0, 0.4], Pd = 92%; Fig. 4*A* and *B*). The difference (boy–girl) in back onset threshold time was 0.6 min (−0.2, 1.4; Pd = 89%; Fig. 4*A*). An age difference of 4 years (from 11 to 15 years) was associated with a change in back onset threshold of 1.1 min (0.1, 2.0; Pd = 96%) for boys and 0.8 min (0.1, 1.5; Pd = 98%) for girls (Fig. 4*A*). The mean difference in back onset threshold time was 0.7 min (−0.3, 1.8; Pd = 88%) between adult males and boys, and 0.5 min (−0.2, 1.3; Pd = 87%) between adult females and girls (Table 3).

The difference (boy-girl) in arm onset threshold time was −0.6 min (−1.2, 0.1; Pd = 91%; Fig. 4*B*). An age difference of 4 years (from 11 to 15 years) was associated with a change in mean arm onset threshold of 0.8 min (−0.1, 1.7; Pd = 92%) for boys and 0.9 min (0.2, 1.6; Pd = 98%) for girls (Fig. 4*B*). The mean difference in arm onset threshold time was 1.4 min (0.5, 2.4; Pd = 99%) between adult males and boys, and −0.5 min (−1.3, 0.1; Pd = 91%) between adult females and girls (Table 3).

## Gastrointestinal temperature

**WARM.** The difference in $\Delta T_{gi}$ between sexes (boys–girls) was −0.04°C (−0.23, 0.14; Pd = 64%). An age difference of 4 years (from 11 to 15 years) resulted in a change in $\Delta T_{gi}$ (younger–older) of 0.00°C (−0.00, 0.00; Pd = 54%) for boys and 0.00°C (−0.00, 0.00; Pd = 96%) for girls. The mean difference in $\Delta T_{gi}$ was 0.01°C (−0.37, 0.33; Pd = 53%) between adult males and boys, and −0.14°C (−0.41, 0.14; Pd = 80%) between adult females and girls (Table 3).

**HOT.** The difference in $\Delta T_{gi}$ between sexes (boys–girls) was −0.14°C (−0.37, 0.08; Pd = 85%). An age difference of 4 years (from 11 to 15 years) resulted in a change in $\Delta T_{gi}$ (younger–older) of 0.00°C (−0.00, 0.00; Pd = 86%) for boys and 0.00°C (−0.00, 0.00; Pd = 99%) for girls. The Mean difference in $\Delta T_{gi}$ was −0.23°C (−0.51, 0.04; Pd = 92%) between adult males and boys, and −0.28°C (−0.53, −0.04; Pd = 97%) between adult females and girls (Table 3).

## Discussion

The aim of this study was to investigate the influence of age and biological sex on the sudomotor response to

**Table 3. Thermoregulatory responses of adults and children exercising in WARM and HOT.**

| Parameter | Boys | Adult males | Girls | Adult females |
|---|---|---|---|---|
| **WARM** | | | | |
| WBSR ($g \cdot h^{-1}$) | 465 | 603 | 473 | 452 |
| | (399, 533) | (557, 649) | (443, 505) | (418, 486) |
| $LSR_{back}$ ($mg \cdot cm^{-2} \cdot min^{-1}$) | 0.80 | 0.97 | 0.83 | 0.78 |
| | (0.72, 0.88) | (0.75, 1.19) | (0.75, 0.90) | (0.51, 1.04) |
| $LSR_{arm}$ ($mg \cdot cm^{-2} \cdot min^{-1}$) | 0.63 | 0.63 | 0.64 | 0.56 |
| | (0.57, 0.69) | (0.50, 0.76) | (0.58, 0.69) | (0.39, 0.72) |
| Back onset (min) | 13.2 | 13.9 | 12.6 | 13.1 |
| | (12.6, 13.8) | (12.9, 14.9) | (12.1, 13.1) | (12.4, 13.8) |
| Arm onset (min) | 13.8 | 15.2 | 14.3 | 15.1 |
| | (13.2, 14.3) | (14.3, 16.1) | (13.9, 14.8) | (13.8, 16.3) |
| $\Delta T_{gi}$ (°C) | 0.83 | 0.82 | 0.87 | 1.01 |
| | (0.69, 0.97) | (0.63, 1.01) | (0.75, 1.00) | (0.76, 1.27) |
| **HOT** | | | | |
| WBSR ($g \cdot h^{-1}$) | 758 | 843 | 702 | 716 |
| | (669, 848) | (742, 943) | (642, 765) | (650, 783) |
| $LSR_{back}$ ($mg \cdot cm^{-2} \cdot min^{-1}$) | 1.17 | 1.09 | 1.09 | 1.01 |
| | (1.05, 1.28) | (0.76, 1.41) | (0.99, 1.19) | (0.72, 1.27) |
| $LSR_{arm}$ ($mg \cdot cm^{-2} \cdot min^{-1}$) | 0.85 | 0.92 | 0.77 | 0.84 |
| | (0.75, 0.94) | (0.76, 1.09) | (0.68, 0.85) | (0.69, 0.99) |
| $\Delta T_{gi}$ (°C) | 1.18 | 0.95 | 1.32 | 1.04 |
| | (1.01, 1.34) | (0.73, 1.16) | (1.16, 1.48) | (0.85, 1.23) |

Values are expressed as the mean (90% credible intervals). Abbreviations: $LSR_{arm}$, end-exercise arm local sweat rate; $LSR_{back}$, end-exercise back local sweat rate; $\Delta T_{gi}$, change in gastrointestinal temperature; WBSR, whole-body sweat rate.

exercise in children, and to compare children with young healthy adults. Our main findings indicate that age was not associated with a meaningful difference in $LSR_{back}$ and $LSR_{arm}$ for boys and girls in WARM. In HOT, a greater age of 4 years (from 11 to 15 years) resulted in a 0.21 and 0.12 $mg \cdot cm^{-2} \cdot min^{-1}$ greater end-exercise $LSR_{back}$ in boys and girls, respectively (Fig. 3*A*). The absence of meaningful differences in LSR between adults and children suggests that the greater LSR of older boys was associated with but not directly influenced by age. WBSR, LSR and onset threshold times did not differ meaningfully between boys and girls, with all between-sex differences being small (e.g. $\leq$0.08 $mg \cdot cm^{-2} \cdot min^{-1}$ for LSR, $\leq$24 $g \cdot h^{-1}$ for WBSR and $\leq$0.6 min for onset threshold) and 90% credible intervals spanning zero, suggesting that biological sex did not implicitly influence sweating in children during exercise.

### Local sweat rate

An age difference of 4 years was associated with a 0.21 $mg \cdot cm^{-2} \cdot min^{-1}$ (−0.49, 0.07; Pd = 93%; Fig. 2*A*) greater end-exercise $LSR_{back}$ of boys exercising in the HOT environment. Previous research has reported a greater sweat output on the torso of young adult males when indirectly compared with prepubertal children

exercising in warm environmental conditions (30°C, 40% RH) (Arlegui et al., 2021). Our findings agree with prior suggestions that older boys and young adult males might have a greater sweat rate on the torso region when compared with prepubertal counterparts (Smith & Havenith, 2011). However, in our study the end-exercise $LSR_{back}$ in adult males was not greater than that in boys in HOT. This finding suggests that the difference in $LSR_{back}$ between younger and older boys might be influenced by factors other than age. For example, aerobic fitness has been suggested to influence sweat distribution, independent of $E_{req}$, with a greater sweating rate of the forehead in highly fit adult males despite no influence on forearm sweat rate (Cramer et al., 2012). Thus, differences in aerobic fitness, or other factors that influence sweat distribution (e.g. heat acclimatization), might have confounded the greater end-exercise $LSR_{back}$ of older boys. Moreover, previous research has demonstrated a large coefficient of variation (18.8%) when investigating the inter-day reliability of LSR in adults, using the absorbent patch technique (Peel et al., 2022). The mean difference in LSR between trials of the same environmental condition was 0.19 $mg \cdot cm^{-2} \cdot min^{-1}$ during exercise at 300 $W \cdot m^{-2}$, which is comparable to the difference in LSR between younger and older children within the present study (0.21 $mg \cdot cm^{-2} \cdot min^{-1}$). Yet, the lower absolute LSR of the children equated to a larger

percentage change (∼21%) when compared with the study by Peel et al. (2022) (∼9%), which might indicate a physiological difference.

In HOT, the apparent association between age and $LSR_{back}$ and $LSR_{arm}$ in girls might reflect the increased WBSR observed in older participants in more thermally stressful conditions. The $E_{req}/E_{max}$ ratio of ∼1.3 indicates that participants experienced uncompensable heat stress, suggesting that age-related differences in sweating emerge primarily under higher heat loads, which elicit greater sweating from the back and forearm. However, if age had a direct effect on sweating, adult females would be expected to exhibit substantially higher LSR than girls. In fact, the differences between adult females and girls were minimal (0.01 mg·cm$^{-2}$·min$^{-1}$ at both sites), indicating that age *per se* probably did not influence LSR in girls, and other factors, such as aerobic fitness or body composition, might have confounded this relationship (Topham et al., 2024b). In the present study, boys and girls did not display meaningful differences in $LSR_{back}$ and $LSR_{arm}$ during exercise in either WARM or HOT environmental conditions (all <0.1 mg·cm$^{-2}$·min$^{-1}$).

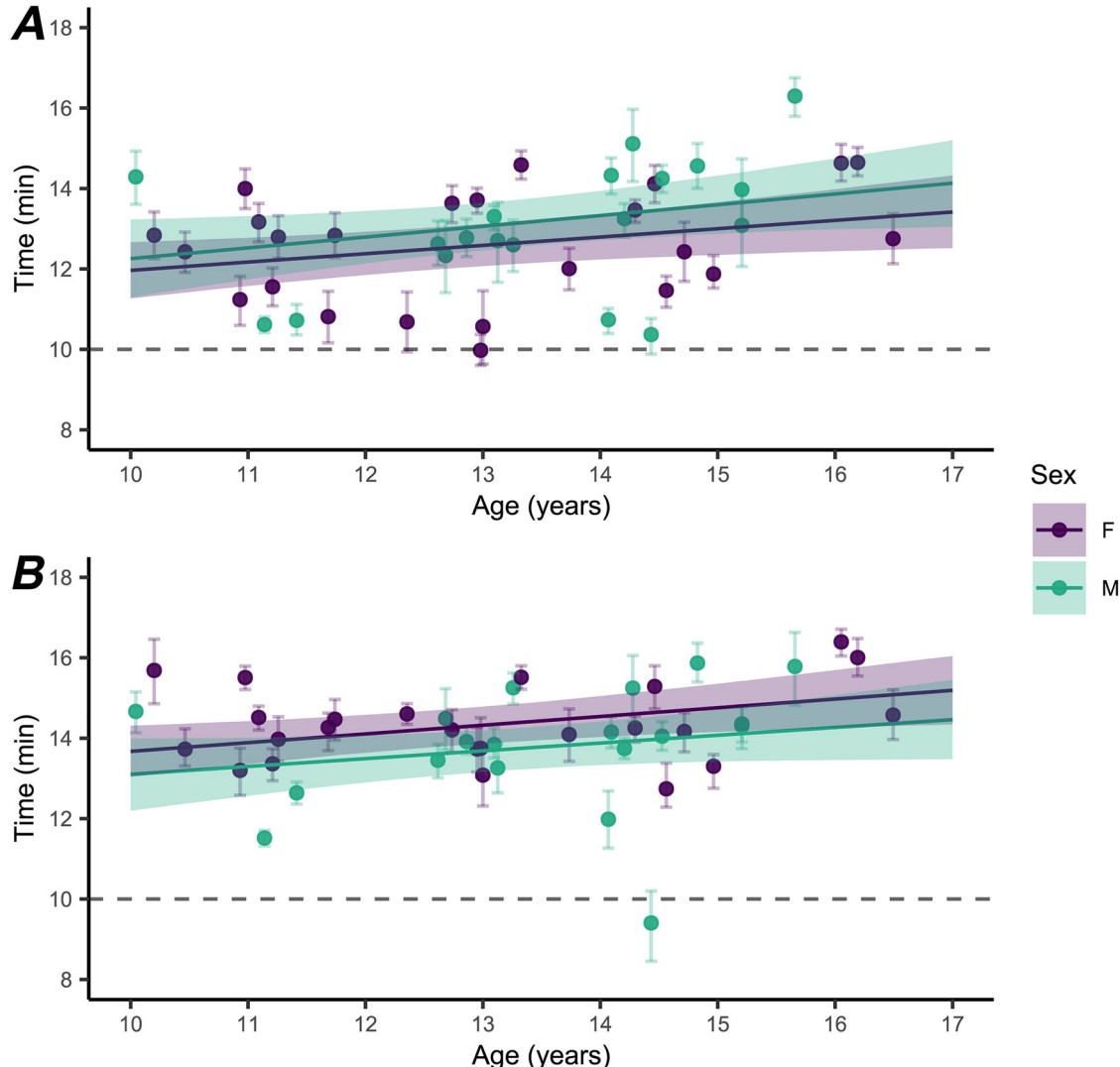

**Figure 4. Time of back onset threshold and arm onset threshold of girls and boys during exercise in WARM conditions relative to age**
Time of back onset threshold (*A*) and arm onset threshold (*B*) of girls (F; *n* = 24) and boys (M; *n* = 20) during exercise in WARM conditions relative to age (circles), with model estimates (line) and 90% credible intervals (CI; ribbon). Error bars represent the 90% CI of model estimates for each individually predicted onset threshold. Data were analyzed with segmented linear regression, with predictor variables as the interaction of age (in years) × sex (male/female). Slopes represent the change in onset threshold time (in minutes) per year of age. Dashed line represents the start of exercise.

Our finding contrasts with a recent report by Amano et al. (2025) that showed distinct differences in LSR of the forearm between girls and boys following pilocarpine iontophoresis to induce localized cholinergic sweating. Differences in BSA between sexes in the study by Amano et al. (2025) might have confounded data interpretation and led to the discrepancy with our results. Indeed, we did not find distinct differences in BSA between boys and girls, whereas Amano et al. (2025) reported differences between 14- and 17-year-olds, but not between 12- and 13-year-olds. The reason for the discrepancy between our study and the 12- and 13-year-olds reported by Amano et al. (2025) is not clear, but if differences in sweating are attributable to the independent effect of biological sex, it is expected that girls would systematically have a lower LSR than boys, because biological sex is categorical. Moreover, Amano et al. (2025) used pilocarpine iontophoresis to stimulate cholinergic sweating, which probably elicited near-maximal sweat gland recruitment, whereas in the present study sweating was induced via exercise in different environmental conditions, driven primarily by $E_{req}$. This fundamental difference in how sweating was evoked might also have contributed to the divergent findings.

## WBSR

In WARM, an age difference of 4 years in boys and girls was associated with a mean increase in WBSR of 60–90 g·h$^{-1}$ (Fig. 3*A*). Furthermore, an increase in age of 4 years in HOT was associated with a 256 and 211 g·h$^{-1}$ increased WBSR for boys and girls, respectively (Fig. 3*B*). Previous research has suggested that advanced sexual maturity is associated with a greater sweat rate in boys (Araki et al., 1979; Meyer & Bar-Or, 1994), but not girls (Rivera-Brown et al., 2006), which has led to the assumption of a male hormonal effect on the sweating response (Inoue et al., 2004). Given that both boys and girls displayed a similar association between age and WBSR in the present study, a male hormonal effect on WBSR does not appear to be responsible for our findings. Furthermore, no meaningful differences in mean WBSR were detected between sexes in both WARM (22 g·h$^{-1}$) and HOT (24 g·h$^{-1}$) environments. Accordingly, it appears that differences in BSA between individuals confounded the relationship between age and WBSR, because the latter is primarily driven by $E_{req}$ in watts, which is correlated with absolute $\dot{H}_{prod}$ (Gagnon et al., 2013; Cramer & Jay, 2014, 2015). In the present study, older children had a greater BSA than their younger counterparts and exercised at a greater absolute $\dot{H}_{prod}$ and $E_{req}$, which would affect WBSR independent of age (Smallcombe et al. 2025).

In HOT, the greatest WBSR occurred in adult males (843 g·h$^{-1}$), who also had the greatest BSA (1.97 m$^2$)

and exercised at the highest absolute $\dot{H}_{prod}$ (598 W; Table 2). Therefore, it does not appear that age mediated an increase in WBSR, but rather that it was attributable to differences in absolute work rate and $E_{req}$. Our data indicate a 30% greater WBSR in adult males than in boys when exercising in WARM. In contrast, the mean difference in WBSR in HOT was 11%. Previous studies isolating the effect of $E_{req}$ on WBSR have done so during exercise in compensable environments (Gagnon et al., 2013; Cramer & Jay, 2014, 2015), because the association between $E_{req}$ and sweat loss is reduced in more uncompensable environments (e.g. HOT), owing to lower sweating efficiency and sweat evaporation contribution. In WARM, the 30% greater WBSR of adult males was comparable to the 27% difference in $E_{req}$ between adult males and boys. In HOT, the 11% difference in WBSR coincided with a 23% difference in $E_{req}$, which was a less positive relationship.

## Sweating onset threshold

In the WARM condition, an increase in age of 4 years was associated with a 1.1 min (0.1–2.0 min) and 0.8 min (0.1–1.5 min) increase in the sweating onset threshold time of boys and girls, respectively (Fig. 4). These results suggest that age was positively associated with time for children to start sweating during exercise at 300 W·m$^{-2}$. The positive association of age with sweating onset threshold time might be attributed to differences in body size between younger and older children. Younger children typically have a lower body mass than older children and, as such, a smaller 'heat sink', which would equate to a faster rise in core temperature and earlier onset of sudomotor activity. Importantly, our results demonstrate that younger children did not have a later onset of sweating when compared with older counterparts. Previous research has reported a 0.8–2.5 min shorter time to sweating onset of the chest, back, forearm and thigh in prepubertal boys compared with adult males during passive heat stress (Shibasaki, Inoue, Kondo, & Iwata, 1997). This finding is comparable to the ˜0.6 min earlier onset of sweating in children when compared with adults in the present study.

To contextualize the sweating responses observed, we compared changes in $\Delta T_{gi}$ between groups. In HOT, adult males had a 0.23°C (−0.04, 0.51) lower $\Delta T_{gi}$ than boys, and adult females had a 0.28°C (0.04, 0.53) lower $\Delta T_{gi}$ than girls. The difference in $\Delta T_{gi}$ can be explained by the differences in $\dot{H}_{prod}$ per unit body mass (in W·kg$^{-1}$) between adults and children (Leites et al., 2016). Given that the exercise intensity was standardized to BSA, mean $\dot{H}_{prod}$ per unit body mass in HOT for adults was ˜8.0 W·kg$^{-1}$, compared with 8.6 and 8.8 W·kg$^{-1}$ for boys and girls, respectively. The $\Delta T_c$ during exercise is

governed primarily by the amount of heat energy stored internally relative to body mass, because the mass of body tissues acts as a heat sink for internal heat gain (Havenith et al., 1998; Cramer & Jay, 2014, 2015). Therefore, it is unsurprising that a greater $\Delta T_{gi}$ occurred in children alongside their higher $\dot{H}_{prod}$ per unit body mass. This finding does not confer that children have a disadvantaged thermoregulatory system when compared with adults, but rather, reflects the greater $\dot{H}_{prod}$ per unit body mass of children when required to exercise at 300 W·m$^{-2}$.

## Limitations

A previous longitudinal study of ˜4000 Australian children reported that 53% of boys and 74% of girls showed signs of puberty at age 11 years (Edwards, 2014). Therefore, we cannot assume that our sample represents prepubertal children. However, our findings are representative of children aged 10–16 years, because we chose a wide spread of ages with the aim of encapsulating pre-, mid- and postpubertal children. A reliable assessment of biological maturation in children is difficult to conduct, owing to the invasive nature of the measurement (i.e. direct measure of genitalia) and the limitation of accurate self-assessed measures of maturational stage (Azevedo et al., 2009). Another limitation of the present study is the specificity of the environmental conditions and measurement sites. Our LSR and onset threshold data were measured and determined at the arm and upper back to represent a site at the extremity and torso, respectively. Different measurement locations might have resulted in different LSR and onset threshold results. However, the absence of notable differences in WBSR between children and adults suggests that LSR and the onset of sweating might not have varied if measured at different locations.

## Conclusion

Our findings indicate that a difference in age of 4 years in both boys and girls did not meaningfully influence LSR and sweating onset threshold when exercising at a fixed $\dot{H}_{prod}$ per BSA in a WARM environment. In contrast, in a HOT environment a difference in age of 4 years elicited a greater LSR. A greater WBSR was also evident in adult males when compared with adult females, boys and girls owing to differences in metabolic heat production mediated by differences in body size. Furthermore, biological sex did not influence sweating in children during exercise in WARM or HOT environments.

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

## Additional information

### Data availability statement

Data are available from the corresponding author (julien.periard@canberra.edu.au) upon reasonable request and signed access agreement.

## Competing interests

Authors have no competing interests to declare.

## Author contributions

T.H.T., J.W.S., B.C., R.D.T., O.J. and J.D.P. conceptualized and designed the research. T.H.T., J.W.S and H.A.B. performed data collection. T.H.T. and A.P.W. performed statistical analyses and created the data visualizations. T.H.T. and J.D.P. interpreted the results and drafted the manuscript. All authors revised and approved the final version of the manuscript and agree to be accountable for all aspects of the work and qualify for authorship on the submitted manuscript.

## Funding

This project was supported by NHMRC Project Grant 2018/GNT1162371 (holders: O. Jay and J. Périard).

## Acknowledgements

The authors thank all participants and the laboratory members at UC and USYD who assisted with data collection.

Open access publishing facilitated by University of Canberra, as part of the Wiley - University of Canberra agreement via the Council of Australasian University Librarians

## Keywords

boys, girls, hot temperature, thermoregulation

## Supporting information

Additional supporting information can be found online in the Supporting Information section at the end of the HTML view of the article. Supporting information files available:

**Peer Review History**

