## [Peer Review History · The Journal of Physiology]

Influence of Age and Biological Sex on Sweating in Children Exercising in Warm and Hot Environments with Comparison to Adults

Thomas H Topham, James W Smallcombe, Harry A Brown, Brad Clark, Andrew P Woodward, Richard D Telford, Ollie Jay, and Julien D Periard

DOI: 10.1113/JP290323

Corresponding author(s): Julien Periard (Julien.Periard@canberra.edu.au)

The following individual(s) involved in review of this submission have agreed to reveal their identity: Davide Filingeri (Referee #1); Nisha Charkoudian (Referee #2)

Review Timeline:

Submission Date:	15-Oct-2025
Editorial Decision:	17-Nov-2025
Revision Received:	24-Feb-2026
Accepted:	11-Mar-2026

Senior Editor: Paul Greenhaff

Reviewing Editor: Zachary Schlader

Transaction Report:

Re: JP-RP-2025-290323 **"Influence of Age and Biological Sex on Sweating in Children Exercising in Warm and Hot Environments, with Comparison to Adults"** by Thomas H Topham, James W Smallcombe, Harry A Brown, Brad Clark, Andrew P Woodward, Richard D Telford, Ollie Jay, and Julien Periard

Dear Dr Periard,

Thank you for submitting your manuscript to The Journal of Physiology. It has been assessed by a Reviewing Editor and by 2 expert referees and we are pleased to tell you that it is potentially acceptable for publication following satisfactory major revision.

Please address all the points raised and incorporate all requested revisions or explain in your Response to Referees why a change has not been made. We hope you will find the comments helpful and that you will be able to return your revised manuscript within 2 months. If your article is NOT for a Special Issue, you may have 9 months to revise. If you require an extension, please contact journal staff: jp@physoc.org. Please note that this letter does not constitute a guarantee for acceptance of your revised manuscript.

REVISION CHECKLIST:

We look forward to receiving your revised submission.

Yours sincerely,

Paul Greenhaff
Senior Editor
The Journal of Physiology

REQUIRED ITEMS FOR REVISION

- Author photo and profile. First or joint first authors are asked to provide a short biography (no more than 100 words for one author or 150 words in total for joint first authors) and a portrait photograph. These should be uploaded and clearly labelled together in a Word document with the revised version of the manuscript. See Information for Authors for further details.
- You must start the Methods section with a paragraph headed Ethical Approval. If experiments were conducted on humans, confirmation that informed consent was obtained, preferably in writing, that the studies conformed to the standards set by the latest revision of the Declaration of Helsinki and that the procedures were approved by a properly constituted ethics committee, which should be named, must be included in the article file. If the research study was registered (clause 35 of the Declaration of Helsinki), the registration database should be indicated, otherwise the lack of registration should be noted as an exception (e.g. The study conformed to the standards set by the Declaration of Helsinki, except for registration in a database). For further information see: <https://physoc.onlinelibrary.wiley.com/hub/human-experiments>.
- Please upload separate high-quality figure files via the submission form.
- Papers must comply with the Statistics Policy: https://jp.msubmit.net/cgi-bin/main.plex?form_type=display_requirements#statistics.

In summary:

- If $n \leq 30$, all data points must be plotted in the figure in a way that reveals their range and distribution. A bar graph with data points overlaid, a box and whisker plot or a violin plot (preferably with data points included) are acceptable formats.
- If $n > 30$, then the entire raw dataset must be made available either as supporting information, or hosted on a not-for-profit repository, e.g. FigShare, with access details provided in the manuscript.
- 'n' clearly defined (e.g. x cells from y slices in z animals) in the Methods. Authors should be mindful of pseudoreplication.
- All relevant 'n' values must be clearly stated in the main text, figures and tables.
- The most appropriate summary statistic (e.g. mean or median and standard deviation) must be used. Standard Error of the Mean (SEM) alone is not permitted.
- Exact p values must be stated. Authors must not use 'greater than' or 'less than'. Exact p values must be stated to three significant figures even when 'no statistical significance' is claimed.

- Please include an Abstract Figure file and an Abstract Figure legend. An appropriate figure legend, which should not exceed 150 words in length, should be included in the main manuscript file. The Abstract Figure is a piece of artwork designed to give readers an immediate understanding of the research and should summarise the main conclusions. If possible, the image should be easily 'readable' from left to right or top to bottom. It should show the physiological relevance of the manuscript so readers can assess the importance and content of its findings. Abstract Figures should not merely recapitulate other figures in the manuscript. Please try to keep the diagram as simple as possible and without superfluous information that may distract from the main conclusion(s). Abstract Figures must be provided by authors no later than the revised manuscript stage and should be uploaded as a separate file during online submission labelled as File Type 'Abstract Figure'. Please also ensure that you include the figure legend in the main article file. All Abstract Figures should be created using BioRender. Authors should use The Journal's premium BioRender account to export high-resolution images. Details on how to use and access the premium account are included as part of this email.

- Please ensure that all figures and tables have a title and legend, and that they have been cited within the main article text.

EDITOR COMMENTS

Reviewing Editor: Ethics Concerns:
None.

Comments for Authors to ensure the paper complies with the Statistics Policy :
Please add n, statistical test information, and statistical inferences to the tables/figures and/or the associated legends.

Comments to the Author:

In my opinion, this is an important study aiming to understand thermoregulation (sweating) between children and adults; a field where there has been much debate (and incorrect dogma). The experimental control and careful study design is a notable highlight. However, I do agree with the reviewers regarding their feedback, particularly:

1) Reviewer #1's points regarding the focus on WBSR vs. LSR. I would also encourage the authors to think about and clearly address the novelty of the rationale for the study, considering the points noted by this reviewer and what is already known in adults (e.g., <https://doi.org/10.1152/jappphysiol.00248.2014>) and children (as noted by this research group: <https://pubmed.ncbi.nlm.nih.gov/40514198/>); this was also highlighted by Reviewer #2.

2) As noted by Reviewer #2, please consider simplifying and presenting the data more clearly to aid interpretation.

I encourage the authors to address these points. Thank you for your submission to J Physiol!

Senior Editor:

Comments to the Author:

Thank you for the manuscript submission to The Journal of Physiology. It has been considered by two reviewers and a reviewing editor. The consensus opinion is that the manuscript describes an important study aimed at understand differences in thermoregulation between children and adults, which is a topic that appears to be subject to debate and therefore of interest to the readership of the Journal. The Reviewing Editor noted that the experimental control is a highlight of the study, but that the reviewers have raised a number of significant concerns that the authors are encouraged to comprehensively address as they impact upon the interpretation of the data.

Regarding specific guidelines of the Journal, if the research study was registered (clause 35 of the Declaration of Helsinki) the registration database should be indicated in the Methods, else the lack of registration should be noted as an exception (e.g. The study conformed to the standards set by the Declaration of Helsinki, except for registration in a database.). The final paragraph of the Methods section should provide full details of the statistical treatment of the data. We look forward to receiving the revised version of the manuscript.

REFEREE COMMENTS

Referee #1:

In this manuscript, the authors aimed to 1) investigate the influence of age and biological sex on local and whole body sweat rates of children (10-16) during exercise at a fixed Hprod per body surface area (W/m^2), and 2) compare the sweating response of children with adults exercising at the same Hprod. The topic has merit and the methodology to investigate the aims pertaining to local sweating responses has value, for which the authors should be commended. The manuscript is mostly well written with limited editorial issues. There are some broader conceptual considerations that I have listed below - especially in the interpretation of the findings and the conclusions - for the authors to review prior to publication.

Main conceptual consideration:

This group has produced valuable research in recent years pertaining to differences in WBSR between children and adults when exercising at intensities scaled to fitness (VO_{2peak}), body mass ($W \cdot kg^{-1}$), and body surface area ($W \cdot m^{-2}$) (i.e. Smallcombe et al., 2025). Also, when considering variations in WBSR, it has been shown that this is "is primarily driven by E_{req} in W , such that any differences in E_{req} between individuals as a product of the exercise intensity prescription, must be standardised before comparing WBSR (Gagnon et al., 112 2013; Cramer & Jay, 2014, 2015)" [Line 109]. Therefore, I am unclear what the value is in highlighting the differences in WBSR between children and adults in this paper, when exercise was prescribed at a fixed Hprod in W/m^2 (which allows for the unbiased investigation of LSR, not WBSR). Yet the main conclusion in the abstract, the final point in your key points, a large section of the discussion and the conclusion relates to the differences in WBSR which have been shown before and are inherent to the study design, which is well set up to investigate LSR. I would therefore suggest that the paper is restructured to maintain its focus so on LSR and sweating onset thresholds, which are the novel findings of this paper, and less so on WBSR. To this point, the first paragraph in the discussion is well written, with a primary focus on the LSR findings which I believe should be reflected throughout the paper.

Methods:

- Statistical analysis: When comparing adults and children, what value is used for the children? Because if there is an associated change in the outcome variable across the 4-year age difference (11-15) then should it not all be investigated as an age continuum? I may have misunderstood the analysis approach, but more clarity may be needed here.

Results:

- Given the investigation of TGI, I would include Hprod in W/kg in Table 2.

Discussion:

- I don't think that TGI needs its own subsection in the discussion. I think it feeds nicely into your discussion of sweating onset thresholds and the potential mechanism for this but again, as the study design was aimed to investigate LSR, there would of course be inherent differences in Hprod in W/kg , thus differences in ΔTGI .

Line 471: The inter-day reliability assessment of LSR measurements by Peel et al. (2022) was done using the absorbent patch technique, not ventilated capsules.

Line 477: Repetition of results seems unnecessary.

Line 492: I would also add that the main comparison to Amano et al. (2025) is the fact that cholinergic stimulation is likely to achieve near maximal recruitment of the glands whereas this study relies on sudomotor drive for sweating.

Line 543: "younger children were not disadvantaged" - Without wider context, the results show younger children have a later sweat onset and a greater increase in core temperature, thus suggesting potential disadvantage. Perhaps add context values of what pushes someone to "thermally disadvantaged" or not.

Line 563: You cite findings that 53% of boys and 74% of girls showed signs of puberty at age 11 (Edwards, 2014) - I would argue that by recruiting 10-16y, you were unlikely to get a sample of pre-pubertal children. So, the limitation is that you are investigating mid to late pubertal and the gap still exists to investigate pre-pubertal.

Referee #2:

The authors present an interesting, novel and important set of data comparing children and adults with regard to thermoregulatory sweating in warm and hot environments. I commend the group on a labor-intensive study with important and novel results. I have a few suggestions for the paper, primarily to improve clarity.

Overall, while I appreciate the statistical rigor of the analysis, the authors are encouraged to speak more plainly in several locations about what the results were and what they mean. Otherwise, for people not familiar with this type of analysis, it is difficult to follow the results section and the meaning of the figures is not entirely clear.

Introduction: The introduction would be improved by a clear statement of (directional) hypotheses regarding effects of aging and sex on the measured responses related to sweating responses. Was there a reason both warm and hot environments were used - did the authors expect different responses between these environments? Also, many would argue that 30 C, 40% RH is also relatively hot - typical summer temperatures in many parts of the world. I would suggest "HOT" and "VERY HOT" for the two conditions.

Another point that would benefit from clarification in the introduction is regarding the difference between the present work and the recently published paper using a similar group of subjects (Smallcombe et al, 2025). As one reviewer, I see the differences, but it might help the emphasis of the present paper if the distinct goals of this work were identified at the outset.

Results: I'm confused by the text explanation of figure 3 A and B. The figure appears to be a correlation, but the text refers to absolute differences and other comparisons that are not obvious in the figure itself. Are these relationships significant? Are they different between the warm and hot environments or between sexes? I don't see where this is explicitly stated.

I find the discussion of sweating onset (lines 409-426 and Figure 4) difficult to interpret. Why did the authors choose to present sweating onset as a time point rather than as a core temperature or mean body temperature? Time is not an independent variable in thermoregulatory physiology - ideally onset threshold would be calculated as a function of internal temperature. This is particularly true since the rate of rise of internal temperature is not going to be the same in all participants, so "10 minutes" probably does not mean the same thermal stimulus at the hypothalamic level in one person as it does in another.

As with Figure 3, please discuss more explicitly the relationships shown in Figure 4. Again, it looks like you are showing correlations but you do not discuss the relationships at all, in terms of whether they are statistically significant and whether they are different (slopes etc) between males and females. It may be that I don't have the statistical expertise to understand your analysis, but I would suggest you explain the "basics" a little more explicitly so your audience can understand what was significant, what was meaningfully different between groups and what was not.

Discussion, line 494 - 503: I would suggest adding here that using pilocarpine iontophoresis is like hitting the area with a sweating stimulus "hammer" and is, by definition, mechanistically different from a physiological, thermoregulatory stimulus such as that used in the present study. Therefore I'm not surprised that the Amano et al study had different results and I don't think this has major bearing on the present study or conclusions.

Minor:

Introduction, line 91: This is a subtle point, but the evaporation of sweat is not a physiological "response" - the sweating itself is the thermoregulatory response. The evaporation of sweat is the physical process by which the skin cooling occurs.

END OF COMMENTS

Responses to Reviewers Comments Document

Dear Editor,

Thank you for the opportunity to revise our manuscript titled “Influence of Age and Biological Sex on Sweating in Children Exercising in Warm and Hot Environments with Comparison to Adults”. Our responses to the reviewers’ comments are presented below. Any changes made in the manuscript are found in red.

On behalf of all the authors, I thank you and the reviewers for your time in reviewing our manuscript and providing constructive feedback to improve its quality.

Kind regards,
Thomas Topham

REQUIRED ITEMS FOR REVISION

- Author photo and profile. First or joint first authors are asked to provide a short biography (no more than 100 words for one author or 150 words in total for joint first authors) and a portrait photograph. These should be uploaded and clearly labelled together in a Word document with the revised version of the manuscript. See Information for Authors for further details.

A: We have uploaded a separate document with this information.

- You must start the Methods section with a paragraph headed Ethical Approval. If experiments were conducted on humans, confirmation that informed consent was obtained, preferably in writing, that the studies conformed to the standards set by the latest revision of the Declaration of Helsinki and that the procedures were approved by a properly constituted ethics committee, which should be named, must be included in the article file. If the research study was registered (clause 35 of the Declaration of Helsinki), the registration database should be indicated, otherwise the lack of registration should be noted as an exception (e.g. The study conformed to the standards set by the Declaration of Helsinki, except for registration in a database). For further information see: <https://physoc.onlinelibrary.wiley.com/hub/human-experiments>.

A: We have edited the methods section by adding the following paragraph:

Ethical approval

Ethical approval was obtained from the Research Ethics committees of UC (#20204538) and USyd (HREC No: 2016/983) and the trials were conducted in accordance with the Declaration of Helsinki, except for registration in a database.

- Please upload separate high-quality figure files via the submission form.

- Papers must comply with the Statistics Policy: https://jp.msubmit.net/cgi-bin/main.plex?form_type=display_requirements#statistics.

In summary:

- If $n \leq 30$, all data points must be plotted in the figure in a way that reveals their range and distribution. A bar graph with data points overlaid, a box and whisker plot or a violin plot (preferably with data points included) are acceptable formats.

- If $n > 30$, then the entire raw dataset must be made available either as supporting information, or hosted on a not-for-profit repository, e.g. FigShare, with access details

provided in the manuscript.

- 'n' clearly defined (e.g. x cells from y slices in z animals) in the Methods. Authors should be mindful of pseudoreplication.
- All relevant 'n' values must be clearly stated in the main text, figures and tables.
- The most appropriate summary statistic (e.g. mean or median and standard deviation) must be used. Standard Error of the Mean (SEM) alone is not permitted.
- Exact p values must be stated. Authors must not use 'greater than' or 'less than'. Exact p values must be stated to three significant figures even when 'no statistical significance' is claimed.

A: Our statistical analyses were conducted within a Bayesian framework to estimate the magnitude and uncertainty of effects rather than to perform null-hypothesis significance testing. Consequently, p-values are not reported, as they are not defined in Bayesian inference. Effects are summarised using posterior means with 90% credible intervals (CrI), which represent the range within which the true value lies with 90% probability given the data and model.

Repeated observations were modelled using hierarchical (multilevel) structures, ensuring that the unit of inference (n) was the independent participant and avoiding pseudo replication. All raw data points are displayed in figures to illustrate distributional characteristics. Data visualisation emphasises raw observations alongside model-derived posterior estimates to ensure transparent interpretation of both the data and the uncertainty in parameter estimates.

- Please include an Abstract Figure file and an Abstract Figure legend. An appropriate figure legend, which should not exceed 150 words in length, should be included in the main manuscript file. The Abstract Figure is a piece of artwork designed to give readers an immediate understanding of the research and should summarise the main conclusions. If possible, the image should be easily 'readable' from left to right or top to bottom. It should show the physiological relevance of the manuscript so readers can assess the importance and content of its findings. Abstract Figures should not merely recapitulate other figures in the manuscript. Please try to keep the diagram as simple as possible and without superfluous information that may distract from the main conclusion(s). Abstract Figures must be provided by authors no later than the revised manuscript stage and should be uploaded as a separate file during online submission labelled as File Type 'Abstract Figure'. Please also ensure that you include the figure legend in the main article file. All Abstract Figures should be created using BioRender. Authors should use The Journal's premium BioRender account to export high-resolution images. Details on how to use and access the premium account are included as part of this email.

- Please ensure that all figures and tables have a title and legend, and that they have been cited within the main article text.

A: We have included an abstract figure. The abstract figure legend has been included after the abstract:

Line 54: “**Abstract Figure.** Local and whole-body sweating responses and sweating onset of boys, girls, adult females, and adult males during treadmill walking at a fixed rate of metabolic heat production per body surface area ($300 \text{ W}\cdot\text{m}^{-2}$) in warm and hot environments. Bayesian hierarchical modelling showed no meaningful sex or age differences in local sweating responses or onset in children.”

All figures and tables have a title, a legend and have been cited within the main text.

EDITOR COMMENTS

Reviewing Editor: Ethics Concerns:

None.

Comments for Authors to ensure the paper complies with the Statistics Policy :
Please add n, statistical test information, and statistical inferences to the tables/figures and/or the associated legends.

Comments to the Author:

In my opinion, this is an important study aiming to understand thermoregulation (sweating) between children and adults; a field where there has been much debate (and incorrect dogma). The experimental control and careful study design is a notable highlight. However, I do agree with the reviewers regarding their feedback, particularly:

1) Reviewer #1's points regarding the focus on WBSR vs. LSR. I would also encourage the authors to think about and clearly address the novelty of the rationale for the study, considering the points noted by this reviewer and what is already known in adults (e.g., <https://doi.org/10.1152/jappphysiol.00248.2014>) and children (as noted by this research group: <https://pubmed.ncbi.nlm.nih.gov/40514198/>); this was also highlighted by Reviewer #2.

A: Thank you for the comment. We have added a sentence within the introduction which clearly addresses the novelty and rationale of the study.

Line 146: “Following our recent investigation of WBSR responses in children and adults (Smallcombe et al., 2025), this study extends that work by providing the first comprehensive examination of age- and biological sex-associated differences in local sweat rate and sweating onset threshold, while controlling for differences in body size via standardisation of metabolic heat production. The primary aim of this study was to investigate the influence of age and biological sex on LSR of children aged 10-16 y exercising in warm and hot environments at a standardised \dot{H}_{prod} per body surface area ($W \cdot m^{-2}$). The secondary aim was to compare the sudomotor response of children with adults exercising at the same target \dot{H}_{prod} to identify child-adult differences in sweating.”

We have edited the manuscript based on the points raised by Reviewer 2 and have provided information in response to Reviewer 2's comments.

2) As noted by Reviewer #2, please consider simplifying and presenting the data more clearly to aid interpretation.

A: In answer to Reviewer 2's suggestion, we have added more information within the figure legends and in text results to aid interpretation.

I encourage the authors to address these points. Thank you for your submission to J Physiol!

Senior Editor:

Comments to the Author:

Thank you for the manuscript submission to The Journal of Physiology. It has been considered by two reviewers and a reviewing editor. The consensus opinion is that the manuscript describes an important study aimed at understand differences in thermoregulation between children and adults, which is a topic that appears to be subject to debate and therefore of interest to the readership of the Journal. The Reviewing Editor noted that the experimental control is a highlight of the study, but that the reviewers have raised a number of significant concerns that the authors are encouraged to comprehensively address as they impact upon the interpretation of the data.

Regarding specific guidelines of the Journal, if the research study was registered (clause 35 of the Declaration of Helsinki) the registration database should be indicated in the Methods, else the lack of registration should be noted as an exception (e.g. The study conformed to the standards set by the Declaration of Helsinki, except for registration in a database.). The final paragraph of the Methods section should provide full details of the statistical treatment of the data. We look forward to receiving the revised version of the manuscript.

A: Thank you for the comment. We have done our upmost to answer the reviewer's comments. We have also edited the methods section by adding the following paragraph:

Ethical approval

Ethical approval was obtained from the Research Ethics committees of UC (#20204538) and USyd (HREC No: 2016/983) and the trials were conducted in accordance with the Declaration of Helsinki, except for registration in a database.

REFeree COMMENTS

Referee #1:

In this manuscript, the authors aimed to 1) investigate the influence of age and biological sex on local and whole body sweat rates of children (10-16) during exercise at a fixed \dot{H}_{prod} per body surface area (W/m^2), and 2) compare the sweating response of children with adults exercising at the same \dot{H}_{prod} . The topic has merit and the methodology to investigate the aims pertaining to local sweating responses has value, for which the authors should be commended. The manuscript is mostly well written with limited editorial issues. There are some broader conceptual considerations that I have listed below - especially in the interpretation of the findings and the conclusions - for the authors to review prior to publication.

Main conceptual consideration:

This group has produced valuable research in recent years pertaining to differences in WBSR between children and adults when exercising at intensities scaled to fitness ($\dot{V}O_{2peak}$), body mass ($W \cdot kg^{-1}$), and body surface area ($W \cdot m^{-2}$) (i.e. Smallcombe et al., 2025). Also, when considering variations in WBSR, it has been shown that this is "is primarily driven by E_{req} in W , such that any differences in E_{req} between individuals as a product of the exercise intensity prescription, must be standardised before comparing WBSR (Gagnon et al., 112 2013; Cramer & Jay, 2014, 2015)" [Line 109]. Therefore, I am unclear what the value is in highlighting the differences in WBSR between children and adults in this paper, when exercise was prescribed at a fixed \dot{H}_{prod} in W/m^2 (which allows for the unbiased investigation of LSR, not WBSR). Yet the main conclusion in the abstract, the final point in your key points, a large section of the discussion and the conclusion relates to the differences in WBSR which have been shown before and are inherent to the study design, which is well set up to investigate LSR. I would therefore suggest that the paper is restructured to maintain its focus so on LSR and sweating onset thresholds, which are the novel findings of this paper, and less so on WBSR. To this point, the first paragraph in the discussion is well written, with a primary focus on the LSR findings which I believe should be reflected throughout the paper.

A: Thank you for the comment. The reason for highlighting the WBSR results was to inform the reader of the physiological pathway mediating the difference in values between cohorts and place the LSR data in context. However, we agree with the reviewer that the emphasis should be more on LSR and have reduced the focus on WBSR throughout the paper. We have made the following changes below:

Edited the first sentence of the abstract, to focus on LSR: “Age and biological sex have been suggested to affect sweating during exercise, but studies that have compared the local sweating response between boys and girls of different ages is limited.”

Removed the concluding sentence of the abstract, and the final key point which focused on WBSR. We have edited the abstract conclusion to focus more on local sweat rate and onset threshold:

Line 50: “In conclusion, age and biological sex did not meaningfully influence the local sweating response, and onset threshold of children and adults exercising at $300 \text{ W} \cdot \text{m}^{-2}$.”

Edited the final paragraph of the introduction to remove the focus on WBSR:

Line 146: “Following our recent investigation of WBSR responses in children and adults (Smallcombe et al., 2025), this study extends that work by providing the first comprehensive examination of age- and biological sex-associated differences in local sweat rate and sweating onset threshold, while controlling for differences in body size via standardisation of metabolic heat production. The primary aim of this study was to investigate the influence of age and biological sex on LSR of children aged 10-16 y exercising in warm and hot environments at a standardised \dot{H}_{prod} per body surface area ($\text{W} \cdot \text{m}^{-2}$). The secondary aim was to compare the sudomotor response of children with adults exercising at the same target \dot{H}_{prod} to identify child-adult differences in sweating.”

Methods:

- Statistical analysis: When comparing adults and children, what value is used for the children? Because if there is an associated change in the outcome variable across the 4-year age difference (11-15) then should it not all be investigated as an age continuum? I may have misunderstood the analysis approach, but more clarity may be needed here.

A: We thank the reviewer for this comment. When comparing adults and children, posterior mean predictions were calculated using the respective mean age for each group (boys and girls). These comparisons were conducted following our analysis within the paediatric cohort, in which age was modelled as a continuum. We have clarified this in the Methods section as follows:

Line 321: “Posterior mean predictions were calculated for adult males, adult females, boys, and girls using their respective mean age.”

Treating all participants on a single age continuum would assume that a given unit change in age has equivalent physiological meaning across childhood and adulthood. For example, a 4-year difference between a 12- and 16-year-old would be treated as statistically equivalent to a 4-year difference between a 20- and 24-year-old, despite well-recognised non-linear maturation-related changes during adolescence. To address this, our analytical approach first examined age-related differences within the paediatric cohort across the age continuum, before performing separate comparisons between children and adults to identify child–adult differences in sudomotor responses, rather than assuming a linear age effect across the full age range.

Results:

- Given the investigation of TGI, I would include Hprod in W/kg in Table 2.

A: We have included Hprod in W/kg within Table 2.

Discussion:

- I don't think that TGI needs its own subsection in the discussion. I think it feeds nicely into your discussion of sweating onset thresholds and the potential mechanism for this but again, as the study design was aimed to investigate LSR, there would of course be inherent differences in Hprod in W/kg, thus differences in ΔT_{gi} .

A: Thank you for your suggestion. We have removed Tgi as its own subsection in the discussion. We have added the following sentence to connect the paragraphs (Line 577): **“To contextualise the sweating responses observed, we compared changes in ΔT_{gi} between groups.”**

Line 471: The inter-day reliability assessment of LSR measurements by Peel et al. (2022) was done using the absorbent patch technique, not ventilated capsules.

A: We have added in the measurement used by Peel et al., to give context to this discussion. However, for clarity, we have added the following on Line 501: **“Moreover, previous research has demonstrated a large coefficient of variation (18.8%) when investigating the inter-day reliability of LSR in adults, using the absorbent patch technique”**

Line 477: Repetition of results seems unnecessary.

A: The paragraph has been edited and the repetition of results removed:

Line 509: **“In HOT, the apparent association between age and LSR_{back} and LSR_{arm} in girls may reflect the increased WBSR observed in older participants under more thermally stressful conditions. The E_{req}/E_{max} ratio of ~ 1.3 indicates that participants experienced uncompensable heat stress, suggesting that age-related differences in sweating emerge primarily under higher heat loads, which elicit greater sweating from the back and forearm. However, if age had a direct effect on sweating, adult females would be expected to exhibit substantially higher LSR than girls. In fact, the differences between adult females and girls were minimal ($0.01 \text{ mg}\cdot\text{cm}^{-2}\cdot\text{min}^{-1}$ at both sites), indicating that age per se likely did not influence LSR in girls, and other factors, such as aerobic fitness or body composition may have confounded this relationship.”**

Line 492: I would also add that the main comparison to Amano et al. (2025) is the fact that cholinergic stimulation is likely to achieve near maximal recruitment of the glands whereas this study relies on sudomotor drive for sweating.

A: Indeed, we have added this to our comparison with the Amano et al. 2025 study:

Line 529: “Moreover, Amano et al. (2025) used pilocarpine iontophoresis to induce cholinergic stimulation, which likely elicited maximal or near-maximal sweat gland recruitment, whereas our study measured exercise induced sweating. This fundamental difference in how sweating was evoked may also contribute to the divergent findings”

Line 543: "younger children were not disadvantaged" - Without wider context, the results show younger children have a later sweat onset and a greater increase in core temperature, thus suggesting potential disadvantage. Perhaps add context values of what pushes someone to "thermally disadvantaged" or not.

A: Thank you for the comment. Table 3 includes the mean onset threshold time, at the arm and back for boys, girls, adult males and adult females. Our results did not show a later onset threshold of children when compared to adults.

We have added information that aids interpretation, in which younger children had an earlier onset threshold than older children (Line 436): “Model estimates indicate a weak positive relationship between onset threshold time and age in both sexes for back (girls: slope = 0.2 min per year [0, 0.4], Pd = 98%; boys: slope = 0.3 min per year [0, 0.5], Pd = 96%), and arm (girls: slope = 0.2 min per year [0.1, 0.4], Pd = 98%; boys: slope = 0.2 min per year [0, 0.4], Pd = 92%; Figure 4A, B).”

Notwithstanding, we have removed the word “disadvantaged” for clarity, and the sentence now reads:

Line 571: “Importantly, our results demonstrate that younger children did not have a later onset of sweating when compared to older counterparts.”

Line 563: You cite findings that 53% of boys and 74% of girls showed signs of puberty at age 11 (Edwards, 2014) - I would argue that by recruiting 10-16y, you were unlikely to get a sample of pre-pubertal children. So, the limitation is that you are investigating mid to late pubertal and the gap still exists to investigate pre-pubertal.

A: Thank you for the comment. We have amended the limitations section:

Line 592: “Therefore, we cannot assume that our sample represents pre-pubertal children.”

It is difficult to conclude that we have or have not investigated pre-pubertal children in our study, due to the difficulties with measuring maturation and the ages at which puberty occurs. However, we do have a representative sample of children aged 10-16 years, and future study is required with children of lower ages (which would have more pre-pubertal children with the sample).

Referee #2:

The authors present an interesting, novel and important set of data comparing children and adults with regard to thermoregulatory sweating in warm and hot environments. I commend the group on a labor-intensive study with important and novel results. I have a few suggestions for the paper, primarily to improve clarity.

Overall, while I appreciate the statistical rigor of the analysis, the authors are encouraged to speak more plainly in several locations about what the results were and what they mean. Otherwise, for people not familiar with this type of analysis, it is difficult to follow the results section and the meaning of the figures is not entirely clear.

Introduction: The introduction would be improved by a clear statement of (directional) hypotheses regarding effects of aging and sex on the measured responses related to sweating responses. Was there a reason both warm and hot environments were used - did the authors expect different responses between these environments? Also, many would argue that 30 C, 40% RH is also relatively hot - typical summer temperatures in many parts of the world. I would suggest "HOT" and "VERY HOT" for the two conditions.

A: We thank the reviewer for these thoughtful and constructive comments. We acknowledge that clearer guidance regarding expected outcomes can improve accessibility for readers less familiar with Bayesian probability modelling. While our statistical approach does not rely on null-hypothesis testing, we agree that outlining directional expectations may aid interpretation.

Accordingly, we have revised the introduction to more explicitly describe the anticipated direction of age- and sex-related differences in sweating, based on existing literature, while avoiding prescriptive hypotheses that could imply linear effects across developmental stages. This approach maintains consistency with our Bayesian framework while improving clarity for the reader.

“Taken together, these findings suggest that biological sex may influence both maximum sweating capacity and sweat distribution in adults, **which is not evident in less thermally stressful environments. If these sex-based differences in sweating were consistent in children, boys would be expected to have a higher sweating capacity compared with girls under conditions of greater thermal strain.** However, whether biological sex has a similar effect within children, and when compared with adults, is unclear.”

Regarding the environmental conditions, both warm and hot environments were deliberately included to examine whether age- and sex-related differences in sweating were consistent across different levels of thermal strain, rather than being specific to a single environmental context. We acknowledge the reviewer’s point that 30°C, 40% RH may be considered hot in

many real-world settings; however, we retained the terminology “warm” and “hot” to reflect relative differences in environmental heat stress within the controlled laboratory context.

Another point that would benefit from clarification in the introduction is regarding the difference between the present work and the recently published paper using a similar group of subjects (Smallcombe et al, 2025). As one reviewer, I see the differences, but it might help the emphasis of the present paper if the distinct goals of this work were identified at the outset.

A: Thank you for this helpful comment. We have revised the introduction to distinguish the aims of the present study from our previously published work (Smallcombe et al., 2025) and to clearly articulate the novelty of this investigation:

Line 146: “Following our recent investigation of WBSR responses in children and adults (Smallcombe et al., 2025), the present study extends that work by providing the first comprehensive examination of age- and biological sex-associated differences in local sweat rate and sweating onset threshold, while controlling for differences in body size via standardisation of metabolic heat production. The primary aim of this study was to investigate the influence of age and biological sex on LSR of children aged 10-16 y exercising in warm and hot environments at a standardised \dot{H}_{prod} per body surface area ($W \cdot m^{-2}$). The secondary aim was to compare the sudomotor response of children with adults exercising at the same target \dot{H}_{prod} to identify child-adult differences in sweating.”

Results: I'm confused by the text explanation of figure 3 A and B. The figure appears to be a correlation, but the text refers to absolute differences and other comparisons that are not obvious in the figure itself. Are these relationships significant? Are they different between the warm and hot environments or between sexes? I don't see where this is explicitly stated.

A: Figure 3A and 3B include individual data points of boys and girls, along with the model estimate for the WBSR of boys and girls. The model estimates the WBSR for an “average” girl and “average” boy using the individual WBSR data points collected. This is very similar to a linear mixed model, but it has been analysed within a Bayesian framework.

Our analyses were conducted using a Bayesian hierarchical framework, which does not rely on p-values for statistical inference. We have included additional information in the figure legends and results text regarding the slope of the relationships. Information contains the slopes of girls and boys along with credible intervals and probability of direction percentages. As this is a linear relationship, the probability of direction of the relationship, is the same for the slope in the figure (per year) and the difference between 11- and 15-year-old children as stated in the results.

Line 410: Per-year changes in WBSR (slope) derived from posterior estimates indicate a weaker relationship in WARM (boys: $14 \text{ g} \cdot \text{h}^{-1}$ per year [-5, 34], $P_d = 88\%$; girls: 24 [8, 40]

$\text{g}\cdot\text{h}^{-1}$ per year, $\text{Pd} = 99\%$) than HOT (boys: $64 \text{ g}\cdot\text{h}^{-1}$ per year [39, 89], $\text{Pd} > 99\%$; girls: $53 \text{ g}\cdot\text{h}^{-1}$ per year [29, 76], $\text{Pd} > 99\%$) with a $< 10 \text{ g}\cdot\text{h}^{-1}$ per year difference between male and female slopes (Figure 3A, B).

“Figure 3. WBSR of girls (F) and boys (M) during exercise in WARM (A) and HOT (B) conditions relative to age (circles) with model estimates (line) and 90% credible intervals (ribbon). WARM: girls: $n = 24$, boys: $n = 20$; HOT: girls: $n = 18$, boys: $n = 17$. Slopes represent the change in hourly WBSR ($\text{g}\cdot\text{h}^{-1}$) per year of age.”

We have made these changes for in text results as well as legends for both figure 3 and 4.

I find the discussion of sweating onset (lines 409-426 and Figure 4) difficult to interpret. Why did the authors choose to present sweating onset as a time point rather than as a core temperature or mean body temperature? Time is not an independent variable in thermoregulatory physiology - ideally onset threshold would be calculated as a function of internal temperature. This is particularly true since the rate of rise of internal temperature is not going to be the same in all participants, so "10 minutes" probably does not mean the same thermal stimulus at the hypothalamic level in one person as it does in another.

A: Thank you for the suggestion. We chose to present sweating onset as a time-based variable rather than a temperature-based threshold because of limitations associated with using gastrointestinal (GI) temperature to define the onset. GI temperature can show a transient dip at the time sweating begins due to redistribution of blood flow away from the GI tract toward the skin. This dip introduces ambiguity when identifying a true inflection point and made it difficult to apply consistent statistical criteria for determining an onset threshold based solely on internal temperature.

To avoid misinterpretation of this artefact, we used time, recorded under tightly standardised exercise conditions, as a stable reference point for comparing the sweating onset threshold across individuals and groups. Importantly, all participants completed the same protocol, allowing the time to onset to be compared reliably across trials. We subsequently analysed GI temperature responses separately to characterise thermal strain without relying on GI temperature to define the onset threshold itself.

As with Figure 3, please discuss more explicitly the relationships shown in Figure 4. Again, it looks like you are showing correlations but you do not discuss the relationships at all, in terms of whether they are statistically significant and whether they are different (slopes etc) between males and females. It may be that I don't have the statistical expertise to understand your analysis, but I would suggest you explain the "basics" a little more explicitly so your audience can understand what was significant, what was meaningfully different between groups and what was not.

A: As highlighted in a previous response, we have included statistical information to the legends of figure 3 and 4. The legend contains the slopes of girls and boys along with

credible intervals and probability of direction percentages. As this is a linear relationship, the probability of direction of the relationship, is the same for the slope in the figure (per year) and the difference between 11- and 15-year-old children as stated in the results.

“Figure 4. Time of back onset threshold (A) and arm onset threshold (B) of girls (F; $n = 24$) and boys (M; $n = 20$) during exercise in WARM conditions relative to age (circles) with model estimates (line) and 90% credible intervals (CI; ribbon). Error bars represent the 90% CI of model estimates for each individually predicted onset threshold. Data analysed with segmented linear regression with predictor variables as the interaction of age (y) \times sex (male/female). Slopes represent the change in onset threshold time (min) per year of age. Dashed line represents the start of exercise.”

In text results:

Line 436: “Model estimates indicate a weak positive relationship between onset threshold time and age in both sexes for back (girls: slope = 0.2 min per year [0, 0.4], Pd = 98%; boys: slope = 0.3 min per year [0, 0.5], Pd = 96%), and arm (girls: slope = 0.2 min per year [0.1, 0.4], Pd = 98%; boys: slope = 0.2 min per year [0, 0.4], Pd = 92%; Figure 4A, B).”

Discussion, line 494 - 503: I would suggest adding here that using pilocarpine iontophoresis is like hitting the area with a sweating stimulus "hammer" and is, by definition, mechanistically different from a physiological, thermoregulatory stimulus such as that used in the present study. Therefore I'm not surprised that the Amano et al study had different results and I don't think this has major bearing on the present study or conclusions.

A: We have added this into our comparison with the Amano et al. 2025 study:

Line 529: “Moreover, Amano et al. (2025) used pilocarpine iontophoresis to stimulate cholinergic sweating, which likely elicited maximal or near-maximal sweat gland recruitment, whereas in the current study sweating was induced via exercise in different environmental conditions, driven primarily by E_{req} . This fundamental difference in how sweating was evoked may also have contributed to the divergent findings.”

Minor:

Introduction, line 91: This is a subtle point, but the evaporation of sweat is not a physiological "response" - the sweating itself is the thermoregulatory response. The evaporation of sweat is the physical process by which the skin cooling occurs.

A: We have edited this sentence within the introduction:

Line 93: “Sweating is an essential thermoregulatory response when exercising under heat stress, with the evaporation of sweat enabling skin cooling”

Dear Professor Periard,

Re: JP-RP-2026-290323R1 **"Influence of Age and Biological Sex on Sweating in Children Exercising in Warm and Hot Environments with Comparison to Adults"** by Thomas H Topham, James W Smallcombe, Harry A Brown, Brad Clark, Andrew P Woodward, Richard D Telford, Ollie Jay, and Julien D Periard

We are pleased to tell you that your paper has been accepted for publication in The Journal of Physiology.

Yours sincerely,

Paul Greenhaff
Senior Editor
The Journal of Physiology

IMPORTANT POINTS TO NOTE FOLLOWING ACCEPTANCE OF YOUR PAPER:

- **IMPORTANT NOTICE ABOUT OPEN ACCESS:** To assist authors whose funding agencies mandate immediate public access to published research findings, The Journal of Physiology allows authors to pay an Open Access (OA) fee to have their papers made freely available immediately on publication.

The Corresponding Author will receive an email from Wiley with details on how to register or log in to Wiley Authors where you will be able to place an order.

- You can check if your funder or institution has a Wiley Open Access Account here:
<https://authors.wiley.com/author-resources/Journal-Authors/open-access/author-compliance-tool.html>

- You can help your research get the attention it deserves! Check out Wiley's free Promotion Guide for best-practice recommendations for promoting your work at: www.wileyauthors.com/eeo/guide. You can learn more about Wiley Editing Services which offers professional video, design, and writing services to create shareable video abstracts, infographics, conference posters, lay summaries, and research news stories for your research at: www.wileyauthors.com/eeo/promotion.

- If you would like to receive our 'Research Roundup', a monthly newsletter highlighting the cutting-edge research published in The Physiological Society's family of journals (The Journal of Physiology, Experimental Physiology, Physiological Reports, The Journal of Nutritional Physiology and The Journal of Precision Medicine: Health and Disease), please click this link, fill in your name and email address and select 'Research Roundup':
<https://www.physoc.org/journals-and-media/membernews>

EDITOR COMMENTS

Reviewing Editor:

Thank you for nicely revising your manuscript. While reviewer #2's remaining concern is valid (and I agree with them), I think you did a nice job of addressing their concern. Congratulations!

Senior Editor:

Thank you for submitting a revised version of the manuscript along with a comprehensive rebuttal document. The reviewers and reviewing editor are of the opinion that the authors have done a good job at revising the manuscript and feel it will be quite influential. Thank you for supporting The Journal of Physiology and congratulations.

REFeree COMMENTS

Referee #1:

I thank the authors again for their clear responses and the improvements made. I would also like to note the following strengths of the revised manuscript:

Impact on the field: The study meaningfully advances paediatric thermoregulation research by clarifying the roles of age and sex when exercise intensity is properly standardised.

Physiological insight: The findings provide valuable understanding, highlighting that metabolic heat production, rather than sex or age, primarily determine sweating responses.

Originality: The combination of a standardised metabolic heat-production protocol with Bayesian hierarchical modelling across child and adult cohorts represents a novel and significant contribution.

Study design and data robustness: The methodological control, detailed measurement techniques, and rigorous statistical approach strengthen confidence in the dataset and its interpretation.

Validity of conclusions: The authors' conclusions are well supported by the evidence and remain appropriately cautious without overextending beyond the data.

Referee #2:

The revised paper by Topham and colleagues is much improved. I continue to submit that the ultimate "meaning" of the analysis is difficult to understand, but it is improved in the revised text - and as I understand the authors' responses there is no additional clarification they can provide with this kind of analysis. I would encourage the authors, in the future, to utilize analyses that can be understood and interpreted in a meaningful way by a large proportion of the physiology and biomedical communities. Otherwise the important research we do (as a community) will not be disseminated as well as it could be.